# Deep learning with satellite images enables high-resolution income estimation: A case study of Buenos Aires

Nicolás F. Abbate[1,2]*, Leonardo Gasparini[2,3],
Franco Ronchetti[4,5], Facundo M. Quiroga[4,5]

**1** Department of Economics, Graduate School of Arts and Science, New York University, New York, New York, United States of America, **2** Centro de Estudios Distributivos, Laborales y Sociales (CEDLAS), IIE-FCE-Universidad Nacional de La Plata, La Plata, Buenos Aires, Argentina, **3** Consejo Nacional de Investigaciones Científicas y Técnicas (CONICET), La Plata, Buenos Aires, Argentina, **4** Instituto de Investigación en Informática LIDI (III-LIDI), Universidad Nacional de La Plata, La Plata, Buenos Aires, Argentina, **5** Comisión de Investigaciones Científicas (CIC-PBA), La Plata, Buenos Aires, Argentina

* abbate.nicolas@nyu.edu

**Editor:** Beata Calka, Military University of Technology Faculty of Civil Engineering and Geodesy: Wojskowa Akademia Techniczna im Jaroslawa Dabrowskiego Wydzial Inzynierii Ladowej i Geodezji, POLAND

## Abstract

High-resolution income data is crucial for informing policy decisions as it allows policymakers to better understand the distribution of wealth and poverty. However, obtaining this information is often cost-prohibitive, especially in developing countries. We evaluate the potential of using high-resolution satellite imagery and machine learning techniques to create income maps with a high level of geographic detail. We train a neural network with satellite images from the Metropolitan Area of Buenos Aires (Argentina) and 2010 census data to estimate per capita income at a 50x50 meter resolution for 2013, 2018 and 2022. The model, based on the EfficientNetV2 architecture, demonstrates strong predictive accuracy for household incomes ($R^2 = 0.878$), achieving a spatial resolution over 20 times finer than existing methods in the literature. The model also allows estimating income maps for arbitrary images, and can therefore be applied at any point in time. Our approach opens up new possibilities for generating highly detailed data, which can be used to assess public policies at a local level, target social programs more effectively, and address information gaps in areas where traditional data collection methods are lacking.

## 1 Introduction

Sociodemographic indicators are essential for the creation and assessment of public policies since they allow us to measure the efficiency of government programs and the population's well-being, all of which improve transparency and accountability. However, in order for these indicators to be used appropriately, they need to be updated and disaggregated. If not updated regularly, the indicators will not be available in time to implement public policies. In addition, if they are not appropriately

**Data availability statement:** The raw satellite imagery underlying this study is proprietary and cannot be shared publicly due to legal restrictions under the end-user license agreements with Airbus DS Geo SA and the Argentine National Commission for Space Activities (CONAE), who has purchased the imagery. However, all derived data, code, and resources required to reproduce the figures, tables, and primary findings of this study are publicly available in a Zenodo repository (DOI: 10.5281/zenodo.11200069), linked as S1 Replication Package (which contains the data required in the Zenodo repo) and S1 Replication Scripts (which contains the scripts in a GitHub repo). For researchers who wish to replicate the full analysis pipeline from the original source imagery, the proprietary Pléiades and Pléiades NEO satellite imagery can be acquired commercially from Airbus through their data portal: https://space-solutions.airbus.com/imagery/. To facilitate this process, we provide the unique product identifiers for each satellite scene used in this study. The identifiers are: Pléiades (2013): DS_PHR1A_201302051411520_FR1_PX_W059S35_0807_03124, DS_PHR1A_201302071357305_FR1_PX_W059S35_0410_06105, DS_PHR1A_201302071357509_FR1_PX_W059S35_0609_05426 Pléiades (2018): DS_PHR1A_201803251356358_FR1_PX_W059S35_0909_03875, DS_PHR1A_201808021356574_FR1_PX_W059S35_0509_06938, DS_PHR1A_201808021357186_FR1_PX_W059S35_0706_06104 Pleiades NEO (2022): 000047717_1_22_STD_A, 000047717_1_24_STD_A, 000047717_1_25_STD_A, 000047717_1_26_STD_A, 000058605_1_3_STD_A, 000058605_1_4_STD_A, 000058605_1_7_STD_A, 000058608_1_2_STD_A. The authors confirm that they did not have any special access privileges to this data and that it is available to any researcher under the same commercial terms.

detailed and disaggregated, they will not properly differentiate the effects of such implemented policies.

Using surveys and census, the traditional data sources for socioeconomic analysis, it is impossible to achieve both disaggregation and high frequency simultaneously. For instance, in Argentina, the main household survey that collects data on unemployment and poverty is published every quarter, and covers only the main 31 urban areas, with spatial disaggregation limited to city-level granularity [1]. On the other hand, censuses provide information at a census tract level. These tracts cover entire blocks in regions with high population density, but they can also include several kilometers in low-density areas. However, in Argentina and various other countries, censuses are conducted every 10 years or so. Moreover, censuses often do not collect information regarding households' incomes or expenses, which hinders the direct estimation of poverty indicators.

High-resolution satellite images have emerged recently as a valuable source of information about welfare, largely due to the advancements in computer vision (CV) algorithms. The use of deep learning models, such as Convolutional Neural Networks (CNN), has enabled us to classify images according to endogenously detected patterns and algorithmically identify objects such as cars, construction areas, roads, crops, and different types of ceilings. The ability to identify these features and objects is crucial as they are closely linked to local wealth and incomes.

Despite these advancements, welfare estimations are still generally constrained to aggregated areas, such as cities or provinces. This contrasts with the progress made in fields like climatology and demography, where similar data inputs have been used to estimate indicators at the grid level. Grid-based estimates have the potential to democratize access to information for a wide range of applications, often extending beyond the original intent of the authors. A notable example is the Gridded Population of the World project [2], which provides global population density estimates in grids approximately 1 km in size. As of 2024, this database has been cited in over 8,000 peer-reviewed articles. However, gridded estimates on income or welfare still cover large areas with grids as large as 2.4 km per side, limiting their usefulness for studying urban phenomena, which often occur at much finer scales [3,4, see].

The main goal of this work is to provide per capita income estimates with a very high level of disaggregation and resolution: a grid with 50$x$50 meter cells, about 20 times finer than previous attempts in the literature. These estimates cover the *Área Metropolitana de Buenos Aires* (from now on, Buenos Aires city) for 2013, 2018, and 2022. Fig 1 provides a map of the study area, showing its population density and the boundaries of the available satellite imagery. To make these estimates, we used small area per capita income estimates based on the 2010 Census and the household survey for the second semester of 2010, along with the unstructured data available in the high-resolution satellite images (0.5 meters per pixel, similar to Google Maps data).

We trained a CNN model from the *EfficientNetV2* family [5]. These models have the distinct advantage of allowing us to identify the key characteristics of the images that best predict income, without making any prior assumptions about which

**Funding:** The author(s) received no specific funding for this work.
**Competing Interests:** The authors have declared that no competing interests exist.

characteristics correlate more with income. This enables better predictive performance compared to simpler models, such as linear regressions based on a set of predefined variables.

The resulting model enables us to make accurate predictions at a census tract level, reaching a $R^2$ of 0.878 over the test set. We assessed the consistency of the developed maps by comparing the results with census and survey data and evaluating case studies of areas with marked income discontinuities, such as informal settlements. In every case, the model's performance is remarkable, surpassing the results from alternative methods used in the literature.

The trained neural network enables the generation of very high-resolution predictions of per capita income, on a grid of 50x50 meter cells. As far as we know, our proposed strategy is the first to produce income estimates solely based on very high-resolution satellite imagery at this level of detail. Compared to existing studies, our estimates offer a 20-fold improvement in resolution (50x50 meters versus 1x1 kilometers), making them far more useful for analyzing policies and trends in densely populated urban areas——where lower-resolution methods fall short. Additionally, since our method relies exclusively on satellite imagery, it enables income estimates over time by utilizing images from different years (specifically 2013, 2018, and 2022 for Buenos Aires). This is particularly useful in urban areas, where income patterns vary significantly both across and within neighborhoods, and where the landscape changes over time. One additional contribution worth mentioning is the way we create the training dataset: by applying *Small Area Estimation* methods [6], we generate high-resolution income estimates even when such data is not available at this granular level. The complete data pipeline is presented in Fig 1, and described in detail in Sect 3.

Although the developed model focuses only on the city of Buenos Aires, the proposed strategy has various applications for academia and policy. Given that our model can predict the average income of an area solely based on a satellite image, it can be used to predict income in areas where no data is available, or where such data is outdated. This could help bridge the information gap in low-income areas where data collection is limited. However, it's essential to consider that the area where the predictions will be made must have similar patterns to the training data, so specific applications require a fine-tuning of this model.

The maps created with this technique could make the development of localized impact evaluations easier. For example, it would be significantly easier to assess the effects of new sewage or tap water piping in the urban development of a specific area, by only making use of the model's predictions before and after the treatment, assuming an appropriate control group is selected.

Improving the temporal and spatial resolution of the maps would also make it easier to evaluate the effectiveness of the targeting of social programs by using geolocated data from the recipients' place of residence. This would reduce dependence on outdated or unreliable administrative data or ad hoc predictions based on surveys. Guessing the income based on a satellite image has the advantage of its transparency, as it is easy to make a visual inspection of the underlying images and intuitively evaluate whether the model predictions are (to some extent) correct or not.

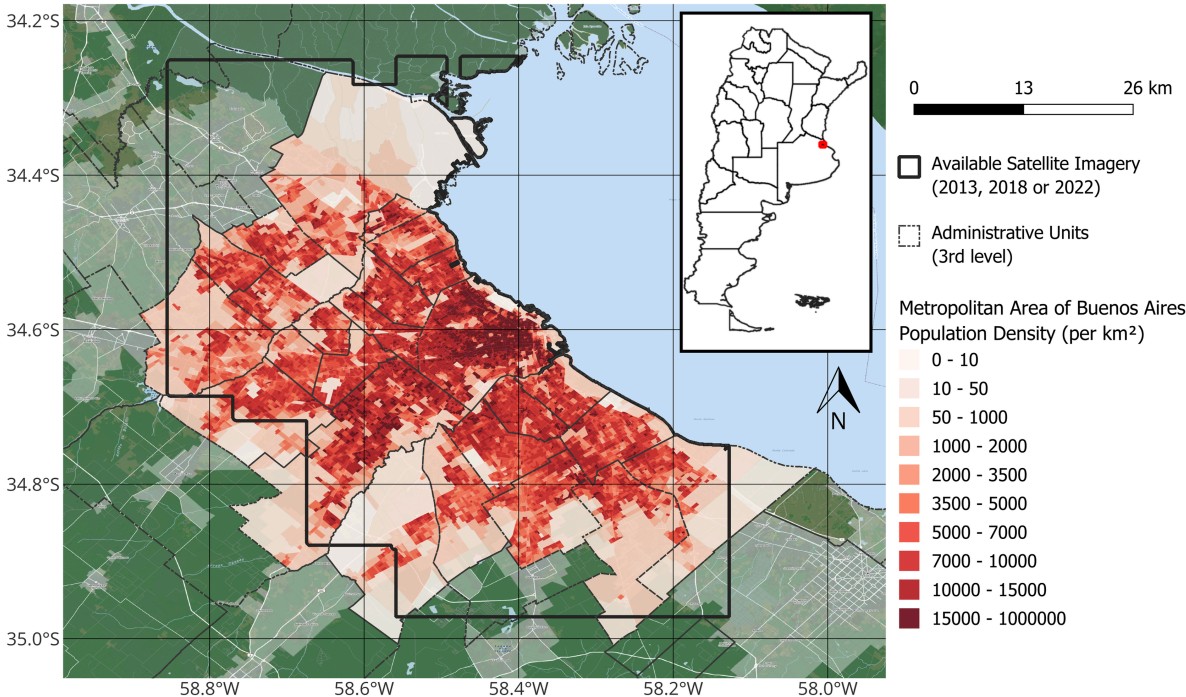

**Fig 1**. **Study area in the metropolitan area of Buenos Aires, Argentina.** The map displays population density (per km²) from the 2010 National Census. The black outline delineates the boundary of the satellite imagery used in the analysis, which is focused on the densely populated urban core, including the Autonomous City of Buenos Aires and its 24 surrounding municipalities (dashed lines).

These techniques could also enhance the targeting of social programs, for example, by using the data for proxy mean tests or as a general exclusion criteria.

In summary, our contributions include:

- The construction of a high-resolution (0.5m) dataset of satellite images in Argentina alongside income estimates of each census tract to train computer vision models.
- The first model to estimate income with 50x50m grids using satellite images, a 20x increase from previous approaches.
- Our CNN based model achieves a surprising $R^2 = 0.878$, better than previous methods.
- The model allows income estimates for Buenos Aires across time (2013, 2018, 2022).
- The method can be replicated to produce urban income estimates in most countries, supporting policy applications in data-scarce areas and localized impact of policy interventions.

This document is organized as follows. In Sect 2 we contextualize this paper into the related literature. In Sect 3, we briefly present the proposed estimation methodology. In the following sections, we depict in detail the methodology used and specify the database creation (Sect 4), the implementation and training of the convolutional neural network, and the procedure used to develop the maps for the years 2013, 2018 and 2022 (Sect 5). The main results of this work are included in Sect 6. Sect 7 conceptualizes the model's measurement of income. Sect 8 discusses the generalization of the model to other cities, including some of its limitations and potential extensions. In Sect 9, we briefly mention how to access the database freely and some recommendations for its use. Sect 10 presents our conclusions and our plans for continuing this line of research.

## 2 Related literature

In recent years, satellite imagery has become a powerful tool for generating estimates across various domains, including agriculture, environmental monitoring, and urban planning. By integrating Machine Learning models with the consistent and global coverage that satellite imagery offers, researchers have been able to develop predictive models for diverse applications such as crop yield estimation, deforestation detection, and urban growth analysis.

However, several challenges have historically hindered the production of income estimates like ours. First, processing the large datasets of high-resolution images required for such estimates can be computationally prohibitive without an efficient data management strategy. Second, very high-resolution satellite imagery is not publicly available; in our case, we accessed images for the city of Buenos Aires through the National Space Commission of Argentina. Finally, for many countries, the absence of disaggregated income data poses a significant obstacle, making it difficult to create a reliable training dataset. Our approach, which incorporates small area estimation techniques to build the training dataset, successfully overcomes these barriers.

This work contributes to the literature on predicting social indicators at a spatial level using *proxies*—observable variables that strongly correlate with the indicator of interest. Notable examples include the use of satellite nighttime lights to predict economic growth [7] and inequality [8]. While income or growth estimates based on nighttime lights share similarities with our approach, one significant limitation of such data is its relatively low spatial resolution compared to daytime reflectance data. For instance, the widely-used VIIRS dataset [9] has a resolution of at least 500 meters near the Equator, which is 100 times lower than the resolution of the images we employ.

An alternative that allows the production of high-resolution estimates of socioeconomic indicators involves the use of geolocalized datasets that closely correlate with income or poverty. Mobile phone data, for example, has been used to predict poverty and wealth [10,11], while social media data has been applied to forecast economic development [12,13]. The main drawback of these methods is that, once trained, they operate on a case-by-case basis and are not easily scalable across multiple countries. This limitation arises because mobile phone and social media usage patterns vary significantly across different social and political contexts, often without a clear correlation with income. Furthermore, access to such data is often restricted, making it challenging for both public and private sectors to obtain. Consequently, while these methods may provide useful estimates in specific cases, their broader applicability is limited.

To address these limitations, daytime high-resolution satellite imagery has gained traction for predicting social indicators [14,15], income [16–18] and wealth [4,19]. Although the performance of these models is generally strong, their predictions typically cover large areas, often at the scale of entire cities or, at best, at a resolution of 1 kilometer [17, see]. Most of these models utilize convolutional neural networks to predict the desired indicators, but recent literature has started adopting Vision Transformers to enhance the models' efficiency [20–23]. A significant downside of these approaches is that they depend on much larger models, making it challenging to replicate this methodology due to the associated computational costs.

The implementation of the model we propose, which combines census and survey information with high-resolution images by applying state-of-the-art convolutional networks, allows us to develop a model with very high accuracy at an unprecedented spatial resolution. In Table 1, we compare our model with similar methodologies used in the literature. The results demonstrate an improvement in the performance of our model compared to similar ones. Our model achieves higher spatial resolution and better performance without requiring additional information beyond what is contained in the satellite images. It's important to note that the models in Table 1 are not directly comparable due to their use of different datasets—some at a national level or from multiple countries, and some using observed data instead of estimated data. Nonetheless, Table 1 provides a clear indication of the effectiveness of our method compared to existing literature, as it produces highly accurate estimates with a spatial scale 20 times finer (50x50 meters compared to 1x1 kilometers).

**Table 1**. Comparison of our model with other strategies.

| | Our Model | Piaggesi et al (2019) | Rolf et al (2021) | Khachiyan et al (2022) | Henderson et al (2012) |
|---|---|---|---|---|---|
| Model | EfficientNetV2 | ResNet50 | Custom CNN | Custom CNN | Linear Model |
| Variable | Per capita income | Per capita income | Per capita income | Total income | Economic growth |
| Images | | | | | |
| Type | Daylight | Daylight | Daylight | Daylight | Nighttime Light |
| Source | Pléiades Neo | DigitalGlobe | Google Maps | LandSat 8 | Suomi NPP |
| Size | 50x50m | 1x1km | 1x1km | 1.2x1.2km | country average |
| Pixels | 128x128px | 224x244px | 256x256px | 128x128px | - |
| $R^2$ | 0.878 | 0.691 | 0.45 | 0.749 | 0.769 |

## 3 Methodology

This study aims to contribute to the emerging field of welfare mapping using satellite images. Our goal is to create a series of maps that estimate household per capita incomes at a sub-municipal level, in cells of approximately 50x50 meters. The estimations are being made for the years 2013, 2018, and 2022. To predict the per capita income at this level of detail, we are utilizing a convolutional neural network. This allows for the extraction of observable features from satellite images that consistently predict the average per capita income of households living in that area. This methodology is based on the proof-of-concept model published in our previous work [24].

Artificial intelligence models, and deep learning in particular, have enabled tackling complex problems that decades of research had not been able to address. Among these, image recognition and classification through these algorithms have sparked a technological revolution [25]. Before the rise of deep learning models, image feature extraction relied on the design and implementation of specialized methods by experts, which required significant effort and resource investment for solutions that only worked on a case-by-case basis. On the contrary, neural networks can identify patterns in images more efficiently and accurately, resulting in better outcomes at a lower cost.

The primary motivation for training a model to predict geographic income from a satellite image of the area of interest arises from the fact that, in many cases, it is possible to visually recognize high, medium, and low-income regions without any additional information. In this sense, we seek to develop a model that can represent the function or mapping between observable features in the images—such as the shape of buildings, materials, paved streets, the presence of green spaces—and the average income of the people living in that area, without defining any functional form, specification, or set of relevant variables a priori. Once trained, the neural network will be able to identify which of these features best predict income endogenously.

To achieve the extraction of features from the images, the model is trained using a series of examples from which it will identify the set of parameters that generate the best predictions. This set of examples, referred to hereafter as the training set, is composed of (i) a collection of images, and (ii) a variable that assigns to each image the indicator to be predicted. In this case, the images we used are satellite captures of the Buenos Aires Metropolitan Area, which includes not only the federal district (usually called the Autonomous City of Buenos Aires, or CABA) but also its metropolitan area. The variable to be predicted is the average per capita income of the households living in the area from which the image was taken. Once the model is trained, it can generate predictions for any set of images, allowing it to be used to predict income at different dates—and therefore produce time series of geographic income—and in other cities with similar characteristics.

The estimation strategy of this work can be broken down into three stages, as shown in Fig 2. Each of these stages is specifically developed in the sections that follow. Step (1), developed in Sect 4, involves constructing a database suitable for training a neural network. This entails estimating the average per capita income by census tract, as this information is not available in a disaggregated form for Argentina. The estimation is performed using a standard small area estimation technique, utilizing a variation of the methodology by *Elbers et al* [6] and *Gasparini et al* [26] to combine data from the Permanent Household Survey for the second semester of 2010 with data from the 2010 National Census of Population,

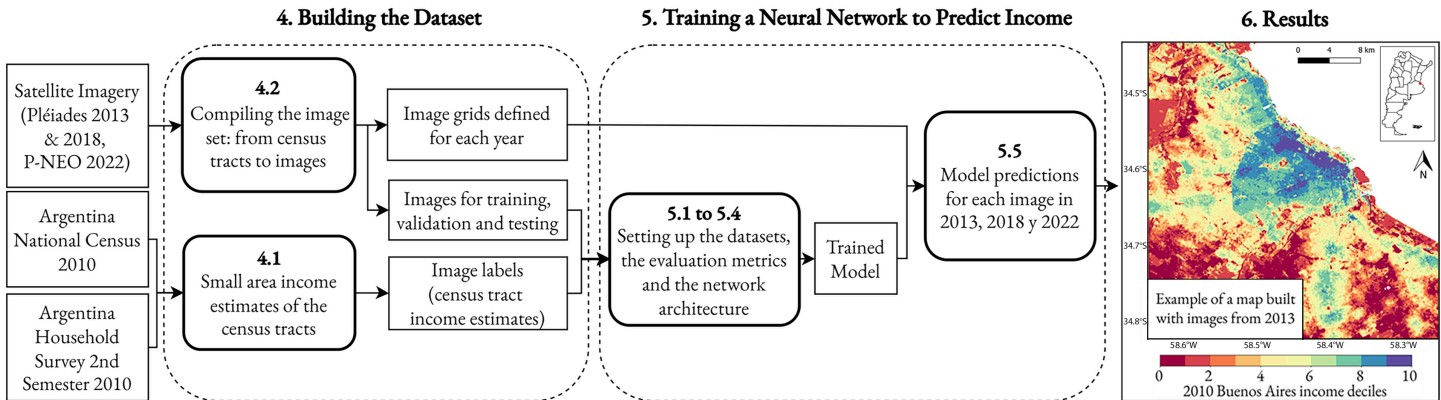

**Fig 2**. **Overview of the estimation strategy by sections of the paper.** This flowchart outlines the three-stage methodology used in the paper. (Sect 4) Building the Dataset: This stage involves creating the training data by combining 2010 census and household survey data through a small area estimation technique to generate income labels for census tracts, which are then paired with high-resolution satellite imagery from 2013. (Sect 5) Training a Neural Network to Predict Income: A Convolutional Neural Network is trained on this dataset to learn the relationship between visual features in the images and per capita income. (Sect 6) Results: The trained model is used to predict income for a high-resolution 50x50 meter grid across the entire study area for multiple years (2013, 2018, 2022), producing detailed income maps.

Households, and Housing. Additionally, a database of satellite images for the Buenos Aires Metropolitan Area in 2013 was constructed, with square images in the format required by the neural network for processing. Step (2), developed in Sect 5, involves training the convolutional neural network (CNN) to relate these satellite images to the estimated income for 2010. Finally, Step (3), developed in Sect 5.5, involves generating a series of income estimates in the form of a 50x50 meter grid for the Buenos Aires Metropolitan Area for the years 2013, 2018, and 2022, using the previously trained model. In other words, a series of highly disaggregated maps is created using the model's predictions for the images from 2013, 2018, and 2022.

The datasets utilized for this study are outlined in Table 2. The two primary household datasets used are the 2010 Census data and the Household Survey from the second semester of 2010, which is the most recent release closest to the census. The satellite image data was provided by the National Commission on Space Activities (CONAE). The decision to utilize satellite images from the years 2013, 2018, and 2022 was based solely on their availability, as CONAE only acquired images for those specific years. While there are publicly accessible alternatives, the resolution of these alternatives is significantly lower compared to the images used in this study. For example, the Sentinel 2 satellite provides free images with a maximum resolution of 10 meters per pixel. In contrast, the images used in this study have a resolution of 0.5 meters per pixel, which is 20 times higher. Lastly, data from the World Settlement Footprint was used to remove low-density areas of Buenos Aires from the training dataset and the city-wide predictions.

## 4 Building the dataset

### 4.1 Creating the labels: Small area income estimates of the census tracts

We estimated the per capita income for each census tract by using microdata from the 2010 Census and the 2010 second-semester household survey data. We followed a derivation of the methodology proposed by *Elbers et al* [6]. While the census provides detailed information about almost every citizen of the country, it does not provide information about household income. The method we implemented allows us to combine spatially disaggregated census data with household survey data, which has a smaller sample size but includes income data of the households.

In their seminal work, *Elbers et al* proposed a small area estimation methodology, which involves: (i) selecting an indicator of interest available in a survey but not in the census, (ii) identifying covariables related to the indicator found in both

**Table 2**. Summary of datasets used in the study.

| Resource | Census data | Household Survey | Satellite Imagery | Population Density Estimates |
|---|---|---|---|---|
| Year | 2010 | 2S 2010 | 2013, 2018 and 2022 | 2015 |
| Type of data | Household Microdata | Household Microdata | Raw TIFF Images | NetCDF |
| Number of samples | 40,117,096 | 118,833 | - | - |
| Storage (GB) | 2.3GB | 0.1GB | 487GB | 0.1GB |
| Source | INDEC[a] | INDEC | CNES[b] (CONAE[c]) | WSF[d] [27]. |

[a] Instituto Nacional de Estadísticas y Censos, Argentina.
[b] Centre national d'études spatiales, France.
[c] Comisión Nacional de Actividades Espaciales, Argentina.
[d] The World Settlement Footprint Dataset.

the census and the survey, (iii) estimating a generalized least squares model that relates the indicator of interest with the selected covariables for the survey data, and (iv) making predictions of the desired indicator using the available census covariables and the model parameters estimated in the previous step. These predictions have the advantage of being geographically localized, as censuses generally have more spatially disaggregated data than household surveys. Typically, after making these predictions, the indicators are aggregated to the minimum spatial level available, which enables the creation of maps with the relevant estimations.

In order to estimate the per capita income for a small area, we used a linear model based on the microdata from the 2010 second-semester household survey (EPH). This model relates the per capita income of each household ($y_i^{eph}$) to a set of observable variables ($X_i^{eph}$), including the gender, education, and age of the head of the household, as well as various factors related to the quality of the house (such as roof and floor materials, access to water and sewage, bathroom facilities, ownership status, and whether it is a precarious building) and the number of household members. Formally the estimated model is:

$$y_i^{eph} = \beta^{eph} X_i^{eph} + \varepsilon_i \tag{1}$$

Once the model is estimated, we fix the $\beta$ parameters vector and generate predictions using these parameters and the census covariables $X_{ij}^c$. In other words, we used the estimated relation from the survey data and used it to predict the income with the census data. Specifically, the following relation is calculated:

$$\hat{y}_{ij} = \beta^{eph} X_{ij}^c \tag{2}$$

which is fairly similar to Eq 1, but instead of estimating the $\beta$ parameters, we use these estimates to make predictions about the income of households in the census based on their observable characteristics. Furthermore, we also label households with a *j* that identifies census tracts, which we will later use to create the income indicator.

Finally, to create the indicator for the census tract average per capita income, we average the predictions of all the households that live in each tract. Thus, after naming the number of households in the *j* census tract $n_j$, the estimated per capita income of such tract ($\hat{Y}_j$) is defined as:

$$\hat{Y}_j = \sum_{i \in j} \frac{\hat{y}_{ij}}{n_j} \tag{3}$$

With such small area estimates, it is possible to create a map of the per capita average income of each census tract. Fig 3 shows the spatial distribution of this indicator in Buenos Aires. The estimated income of each tract is the main input, along with the satellite images, which we use to train the computer vision model.

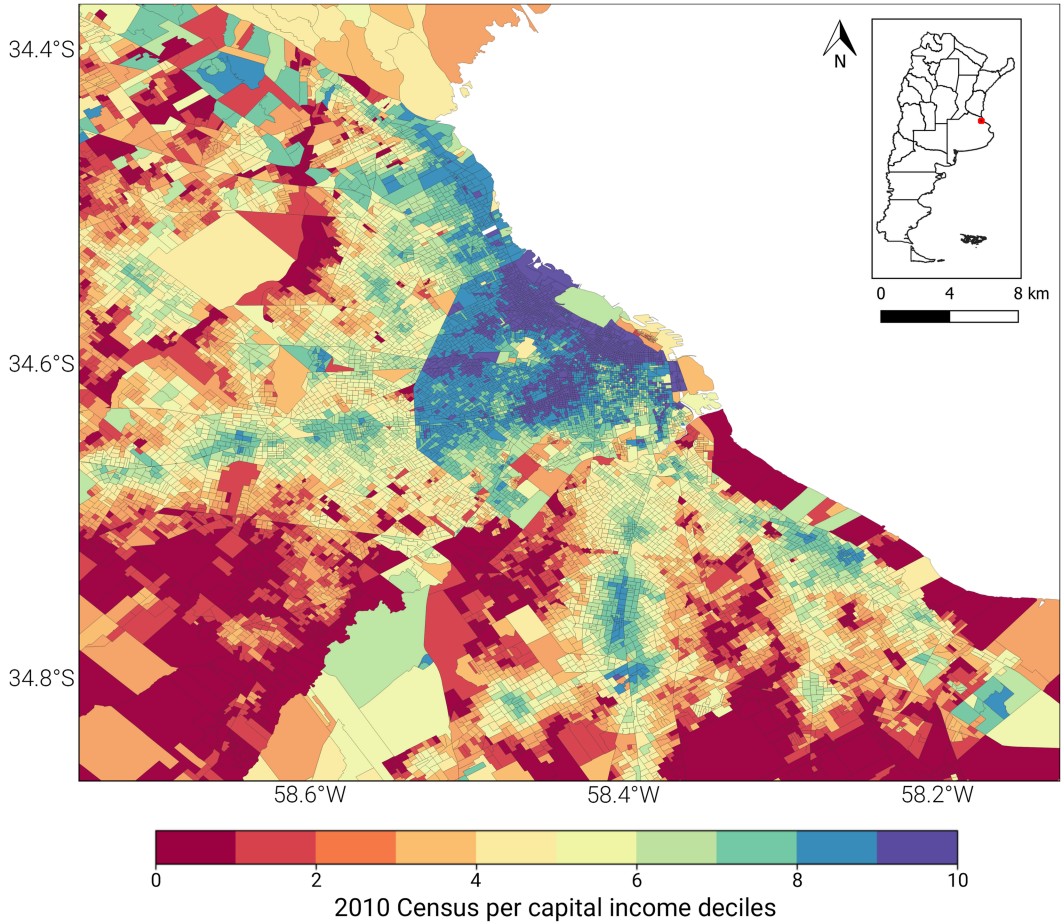

**Fig 3**. **Spatial distribution of per capita average income estimated for Buenos Aires at census tract level.** This map of the Buenos Aires Metropolitan Area shows the spatial distribution of the estimated per capita income used as the "ground truth" for training the model. Each colored polygon represents a census tract, with its color corresponding to the decile of average household' income for 2010. This data was generated using a small area estimation method that combines microdata from the 2010 National Census and the Permanent Household Survey.

In contrast to the approach suggested by *Elbers et al* [6], where estimations were made using generalized least squares (GLS), we opted for an ordinary least squares model due to differences in the available data. The original study had access to a survey that identified households with the census minimum aggregation unit. However, Argentina's survey data does not include information about the specific census tract to which the household data belongs. Consequently, it is not feasible to estimate the covariance matrix required for a GLS estimation.

Despite these difficulties regarding the quality of the original information, we expect the results of these estimations to have only a moderate bias. Although each household estimation will be biased, we expect the aggregation of income per census tract to mitigate this variation to a certain extent, since errors are not perfectly correlated and would counteract each other to a great extent. For example, *Gasparini et al* [26] resort to a similar implementation for poverty estimations throughout time.

In any case, while the estimated indicators may have small biases and errors, the goal of this initial study is centered on evaluating the possibility of using computer vision models to perform income estimations. The application of these models in geographical income estimations is not dependent on the validity of the proposed small area estimation technique. They can also be applied to any type of geographical dataset, as long as the images have distinct features that

strongly correlate with the outcome of interest. Furthermore, any improvements in the quality of the indicator estimates would enhance the final results of the AI algorithm.

## 4.2 Compiling the image set: From census tracts to images

The satellite images used are daytime reflectance images from the Earth's surface taken by the Pléiades satellite constellation, from the Centre National d'Etudes Spatiales (CNES). This constellation produces images of around 0.5 meters per pixel resolution with a revisiting frequency of 26 days. Although the images are not available for the general public, the Argentine Commission of Space Activities (CONAE) provided us access to them. Many governments have access to similar high-resolution satellite images through existing partnerships with imagery providers, frequently utilizing them for a range of applications beyond AI-based predictions, including urban management, environmental monitoring, and disaster response. In comparison, the publicly available satellite images of higher resolution are the ones taken by the Landsat 8 satellite; they have a resolution 30 times lower than Pléiades images (15 meters per pixel).

We utilized a series of images captured in Buenos Aires on February 5th and 7th, 2013, which are the closest available to the census conducted on October 27th, 2010. To ensure a diverse training dataset, we also incorporated images from 2018 and 2022. Although including images further away from the census might pose a bias (since we will show images from, for example, 2022, with input labels from 2010), the aim of including these images is to enable a better generalization of the model, if we assume there are no great differences between the images from different years. In other words, we assume that urban changes take time and, thus, the bias when showing 10-year-apart images is low. This bias-generalization trade-off is assessed empirically in the Results Sect 6 by comparing the model performance on 2010 test images. Interestingly, including "future" images from Buenos Aires allows us to improve the model's performance and to get a higher spatial consistency when predicting in those three periods.

When we did not include the images from those three years and carried out the training with only the 2013 images, the future predictions for 2018 and 2022 are far more volatile, possibly due to light, satellite view angle and vegetation differences, which the model interprets as relevant features. For example, when we only used images from 2013 for training, and then combined images from 2013, 2018, and 2022 for the final model, the model consistently predicted lower incomes when analyzing the 2018 images. This was due to the drier month and yellower vegetation depicted in the images, which are correlated with lower-income households. Similarly, an income increase was perceived in the 2022 images due to the evening time in which they were taken, which cast shadows on the buildings, indicating the presence of taller buildings and, therefore, higher incomes. These types of factors could have been addressed by the model with more diverse examples, highlighting the need to use a greater number of images from different points in time for future developments of these models.

The images comprise four spectral bands representing the colors blue (430 - 550 nm), green (500 - 620 nm), red (590 - 710 nm), and the near-infrared region (NIR) (740 - 940 nm). Additionally, the panchromatic band (470 - 830 nm) is also available. We enhanced the color histogram and used the Brovey transformation (Vrabel, 1996) to increase the resolution of the original images from 1 meter per pixel (the red, green, blue, and NIR bands' original resolution) to 0.5 meters per pixel (the panchromatic band resolution). This process, known as "pansharpening", involves combining the panchromatic band with the lower resolution spectral bands to match the resolution of the panchromatic bands. The resulting images are clearer and more easily detectable for the computer vision model.

One of the main challenges when implementing an image processing model, such as the one proposed, is determining the appropriate spatial dimension of the inputs of the model. CNNs require all images in the dataset to be rectangular and have the same size in pixels. However, as illustrated in Fig 3, the census tracts vary in size, as they are selected to contain a similar number of households. Therefore, as the population density decreases, the size of the census tracts grows. Consequently, it is necessary to preprocess the datasets to ensure that the geographical information has a uniform size.

The image size is not an obvious decision *a priori*. On the one hand, including a higher resolution than that from the original data can bring a significant improvement in the resolution of the created maps, despite a possible quality loss of those estimations. If the images used are too small (say 10*x*10 meters), the results will likely be unsatisfactory since many relevant characteristics for the measurement of income will exceed the image size.

This trade-off is assessed in the Results Sect 6 by comparing the models' performance with images of 50*x*50, 100*x*100 and 200*x*200 meter images, the last size being the most similar to that from many census tracts. The pixel size always stays at 128*x*128px to guarantee that the results are comparable. In addition, we assess a strategy that could allow us to solve the scale problem: combining images with a reduced size (for example, 50*x*50m) with images with a bigger size (for example, 200*x*200m). In other words, instead of using a 4-band image, we build an 8-band stacked composition where the first 4 correspond to a smaller size image and the next 4, to a bigger one.

This stacking approach significantly improves the spatial coherence of predictions and reduces noise. As detailed in the Supplementary Material (S1 Figure and S1 Table), models trained with stacked images, particularly the 50*x*50 + 200*x*200m configuration, produce income maps with substantially higher spatial autocorrelation (measured by Moran's I [28]) compared to models using unstacked images. This increased spatial smoothness more closely mirrors the characteristics of our training data and leads to visually less noisy and more interpretable income maps, alongside achieving a lower Mean Squared Error (see Table 3). The larger contextual window provided by the stacked images likely helps the model disambiguate local features by considering broader neighborhood characteristics, thus stabilizing predictions for the central 50*x*50m area.

Fig 4 provides an example of this stacked image composition for a randomly selected image from Buenos Aires. Combining information from small areas with higher resolution and big areas with lower resolution allows the model to process the information of both images together, without the need to modify the implemented architecture. This could improve the predictions by considering the income-relevant factors that may not be included in the near image. For instance, if we compare two neighborhoods with similar buildings and one of them is near a freeway, it is likely that we are talking about a lower-income household than in the opposite case.

During model training, we dynamically sample the dataset by selecting a random selection of 5 images from each census tract in each training epoch, for a total of about 35000 images per epoch. To construct these images, we select a random point within the tract and create a square of size 50*x*50 (and 200*x*200) meters, centered at the selected point. This means that many images will contain partial data about adjacent tracts, which might have valuable information, particularly when there are discrete income variations or relevant characteristics such as freeways or rivers. Selecting a constant number of images for each census tract, instead of building a grid for the whole Buenos Aires area, allows us to avoid overrepresentation of low population density images in the training process. At the same time, since the images from each period are randomized, the model constantly faces slightly different images. Also, we applied other standard data augmentation techniques by randomly modifying the brightness, contrast, and orientation of the images.

We label each image with the average per capita income estimates of the census tract (see Sect 4.1). We made this decision because there is a strong spatial correlation in the per capita income map, and images from areas and neighborhoods with similar incomes were also quite similar. Although there are significant variations within some census tracts, in general, these tend to be visually homogeneous. Therefore, the average income of the tract is usually a valid estimate of the income of a subgroup of that census tract.

Fig 5 shows an example of two census tracts from which we selected 4 images from each. We only show 4 for convenience, but the actual dataset generates 5 images per census tract. On the left, we present two census tracts with different levels of income. On the right, we show the 4 generated images, similar to the ones used to feed the model, along with their labels in the upper left corner of the image.

## Raw Images (RGB+NIR)

50x50 meters

200x200 meters

## Stacked Images (RGB+NIR): 50x50 + 200x200

50x50 meters

200x200 meters

**Fig 4**. **Multi-resolution approach for generating the stack of images used as input for our model.** The lower-resolution images provide context for the prediction on the 50x50m area. This diagram illustrates the multi-resolution image stacking approach used as input for the CNN model. To capture both detailed local features and broader contextual information, a high-resolution satellite image with dimensions of 50x50 meters is combined with a lower-resolution image of 200x200 meters from the same central area. This stacked input significantly enhances prediction accuracy and generates more spatially coherent income maps. The images displayed are aerial photographs from the Instituto Geográfico Nacional de la República Argentina (2025), provided for illustrative purposes, as the underlying satellite data is subject to licensing restrictions.

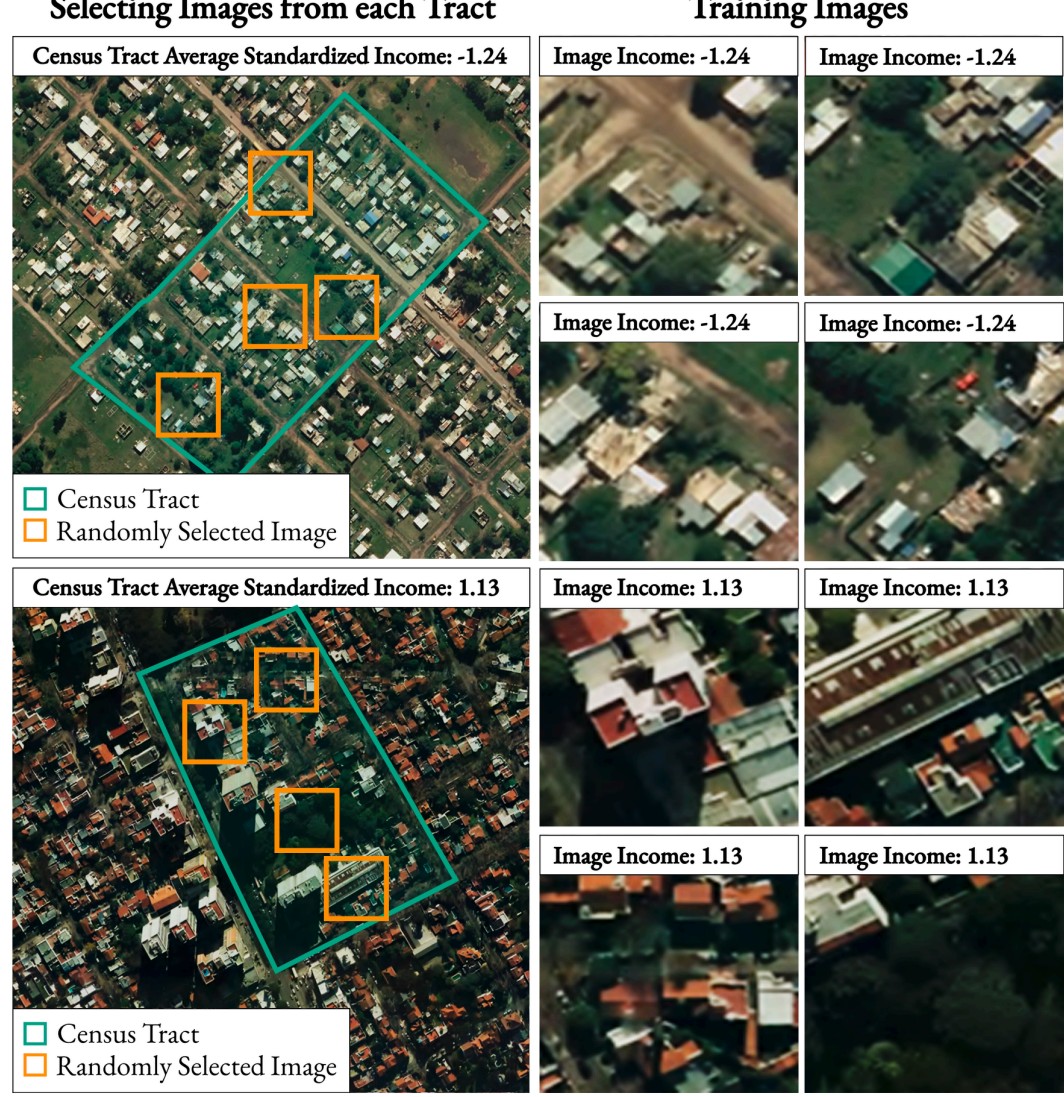

**Selecting Images from each Tract**

**Training Images**

**Fig 5**. **Example of images generated from census tract data, 128x128px (50x50 meters).** This figure demonstrates the process of generating image samples for model training. On the left, two census tracts with different average income levels are shown. On the right, several 50x50 meter square images are depicted, which are randomly sampled from within these tracts. Each of these smaller images is assigned the average per capita income of its respective census tract, which serves as the training label. The images shown are aerial imagery (from *Instituto Geográfico Nacional de la República Argentina*, 2025) for illustrative purposes as the underlying satellite data has licensing restrictions.

## 5 Training a neural network to predict income

CNNs have a significant advantage over other methods in the literature, such as linear models, because they can automatically detect important features in the input data without human intervention. We aim to train a network that extracts relevant latent economic information available in satellite images that correlates with per capita income. The distribution of the observable characteristics contained in satellite images varies a lot within the same city: more vegetation and land in suburban areas; more asphalt and cement near freeways; houses built with sheet metal, and dirt roads in informal settlements. The shapes of the objects also varies, showing houses with broad gardens and green areas in high and middle-income suburbs, and interconnected and compact drainage infrastructure in urban centers [16,29,30].

In the section that follows, we briefly discuss why a CNN is employed to predict income based on features available in satellite images. In the following subsections, we describe the main inputs required to train a CNN, such as the Training, Validation, and Test sets (Sect 5.2), evaluation metrics (Sect 5.4) and the architecture and the hyperparameters set (Sect 5.4).

## 5.1 Why use a convolutional neural network?

Neural Networks consist of a sequence of layers in which a parameterized nonlinear transformation of their inputs—the images—is implemented to generate a prediction—the per capita income of households living in that area. The number of parameters in the model can be very large; for example, the architecture used in this study, called EfficientNetV2S, has 21.6 million parameters [5]. The parameters are generally initialized randomly. During training, the parameters are adjusted to improve the accuracy of the predictions over the training set. In machine learning literature, this method is known as supervised learning [31]. Supervised learning is currently the most widespread paradigm in artificial intelligence and serves as the basis for the vast majority of image processing models. In the case of image processing, supervised learning requires (i) a large set of images and (ii) a value to predict for each image. In this study we use satellite images of Buenos Aires, along with the logarithm of the average per capita income of the households living in the area corresponding to those images, generated according to the estimation method described in the previous section.

Supervised learning involves using the mentioned databases to adjust the model's parameters so that the error in the predictions is minimized. During training, the algorithm is iteratively presented with the images from the training set, and it produces a prediction of the indicator for each one. The goal is for the predicted value to be as close as possible to the actual value of the explained variable. To measure this difference between the predicted and actual values, a loss function is computed to quantify these errors. The algorithm then seeks to minimize the loss function by iteratively adjusting its parameters. To achieve an appropriate adjustment of the parameter vector, the algorithm calculates a gradient vector, which indicates for each parameter how much it should be adjusted to reduce the error. The parameter vector is then adjusted in the opposite direction of the gradient vector. In practice, the typical way to solve this type of problem involves adjusting the parameters iteratively, computing the average gradient for a small sample of examples in each iteration. Each complete iteration over the dataset is called an "epoch". A model may require a significant number of epochs for training, sometimes reaching hundreds. Finally, once an optimal parameter vector is found, the algorithm's generalization ability is evaluated on a new set, typically called the test set.

Among artificial intelligence algorithms, convolutional neural networks (CNNs) are particularly noteworthy due to their widespread use in image processing. In these algorithms, the inputs to the first layer are the images as a matrix. The output of the first layer is used as input for the second layer, and so on. The transformation implemented by each layer is typically a convolution or pooling operation [32], although more recent models like the one used here present more complex transformations [5]. The output of each layer is another "image," with lower resolution than the original, where the pixels represent a summary of the information from the previous layer; in other words, each layer extracts the main features of the previous image. Gradually, the model reduces the size of the image and decreases the resolution of the feature map until finally relating it to the variable to be predicted.

It is important to note that CNNs are robust to "noisy" data during training; that is, if the errors in the target variable are random, CNNs can generalize the underlying function appropriately [33]. This property is relevant because, as mentioned in Sect 4.2, the variable used will not correspond unequivocally to each of the images; instead, in all cases, we will have images with a repeated income value. This is because 5 different images will be selected from each census tract—some from wealthier parts of the tract, others from poorer parts—all with the same variable value: the average income of the census tract. The ability to generalize the model in the presence of these challenges is crucial. However, it should be noted that the algorithm's learning capacity does depend on the validity of the input data: if there are systematic biases in the estimation of per capita income by census tract, the model will replicate those biases in its final results. Therefore, it is highly important to develop techniques that improve the accuracy and validity of small area estimates.

## 5.2 Splitting into training, validation, and test sets

Neural network models generally tend to overfit, meaning they not only learn the general characteristics of the training set but also memorize specific characteristics, which limits their ability to generalize to other data sets. In this work, we aim to develop a model to estimate income based on images from various areas in Buenos Aires, spanning different years and urban characteristics. Therefore, we need a model that can identify and generalize patterns beyond the training set.

Thus, we follow the standard procedure of using three disjoint subsets of training, validation and test. The division will be made at the census tract level, ensuring that all images from a specific tract are included in the same subset. The training set comprises approximately 75% of the census tracts of the city of Buenos Aires (about 7000 census tracts), the validation set 5% of the census tracts (about 500), and the test set, 20% (about 2000). Low population density census tracts are excluded from the analysis.

The allocation criteria of each tract to each set are shown in Fig 6. To construct the test set, we selected two 0.05-degree wide strips (around 4.5km) that cross Buenos Aires from north to south and which include 25% of the census tracts. Specifically, we selected the area between the longitudes –58.71 and –58.66 and between –58.41 and –58.36. The idea here was to ensure that we had an appropriate variability in terms of the income, while avoiding any potential cross-contamination between images of the train/validation set with the test set. We discarded those images that were crossed by these longitudes to avoid this cross-contamination. To define the validation set, we randomly allocated the census tracts that were not used for the test set to the training and validation sets, to achieve a distribution of 70% census tracts in the training set, 5% in the validation set, and the remaining 25% tracts to the test set.

## 5.3 Evaluation metrics

Once the census tracts belonging to the test set and the image selection process are established, it is important to determine the mechanism by which we choose the "best" model among the different architectures used and within the same training process for a specific architecture.

We use the Mean Squared Error (MSE) as the main evaluation metric to train and evaluate the model. During the training phase, we used this metric as a loss function to compare the predicted income of each image with the average income of the census tract ($MSE^{train}$). However, for model evaluation, we used the MSE computed at a census tract level rather than at an image level ($MSE^{test}$). Calculating the MSE directly for each image would overestimate the real error, as part of the difference between the prediction and the income of that tract can be caused by variations within the same tract. In other words, a model that correctly predicts income based on images would have a high MSE in many census tracts since the prediction is related to a subgroup of the tract (the area delimited by the image) and not the average income of the entire tract. Instead, we propose to compute the MSE of the average of the predictions in each census tract. This way, we would be comparing the average predicted income across all the tract images with the average income of each census tract. To simplify this comparison, in the results section, we estimate the $R`2$, which normalizes the MSE by dividing it by the variance, creating an indicator that goes from 0 to 1, 0 being a model with zero prediction power (MSE equal to the variance) and 1, a model that predicts the indicator perfectly (MSE equal to 0).

Formally, we define the MSE of every image as:

$$MSE^{train} = \frac{1}{n_j n_h} \sum_{j \in J_T} \sum_{h \in H_j} \left( \hat{p}_{hj} - \hat{Y}_j \right)^2 \tag{4}$$

where $\hat{p}_{hj}$ represents the prediction of the model for image h of the census tract $j$, $\hat{Y}_j$ represents the census tract per capita income, $J_T$ represents the set of census tracts that belong to the test set and $H_j$ represents the set of images that belong to the $j$ census tract.

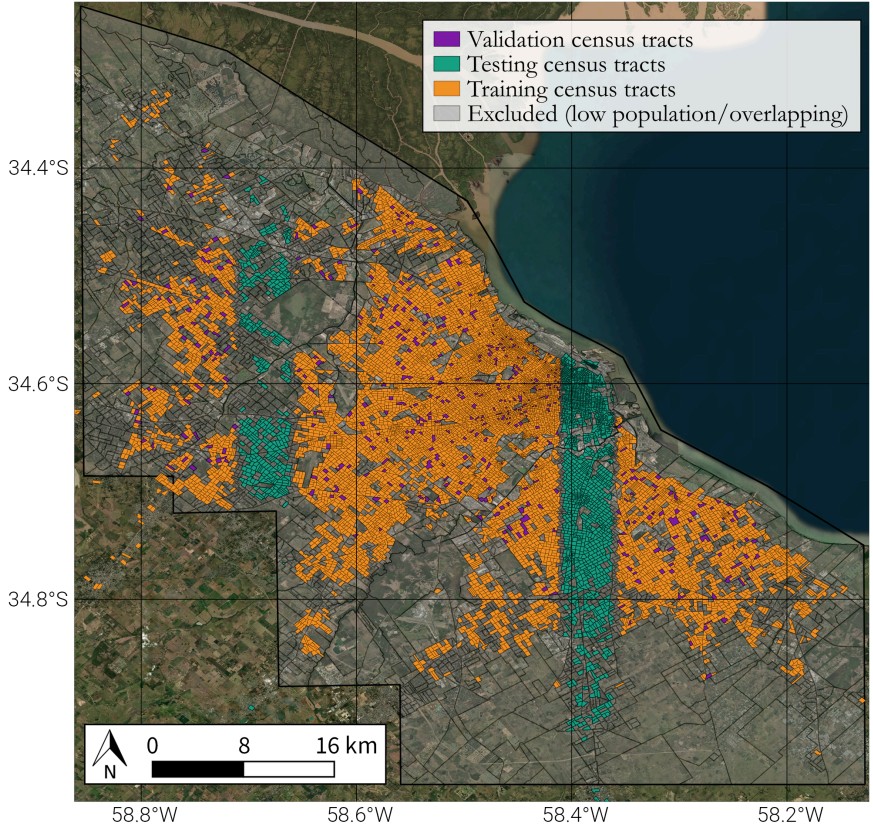

**Fig 6**. **Assignment of census tracts to training, validation and test sets.** This map of Buenos Aires illustrates how census tracts were partitioned to prevent data leakage and ensure a robust model evaluation. Two vertical geographic strips (green) were designated as the test set. The remaining tracts were randomly divided into a training set (orange, 70% of the total) and a validation set (purple, 5%). This spatial separation guarantees that the model is tested on geographically distinct areas it has not seen during training. Background satellite imagery comes from USGS/NASA Landsat.

Following this logic, in a correctly estimated model, $\hat{p}_{hj}$ should have a positive variance within the $j$ census tract, while $\hat{Y}_j$ is constant within the same tract. Therefore, the evaluation metric underestimates the model's precision. To obtain an appropriate MSE, instead of comparing each image, it is possible to compare the prediction average of every image from the tract with the per capita income according to the households' information. Taking this into account, the MSE we use to assess the model's performance is:

$$MSE^{test} = \frac{1}{n_j} \sum_{j \in J_T} \left( \hat{P}_j - \hat{Y}_j \right)^2 \qquad (5)$$

computed for every census tract $j$ that belongs to the $J_T$ test subset. In this expression, $\hat{P}_j$ represents the prediction average of every image in the j census tract:

$$\hat{P}_j = \frac{1}{n_h} \sum_{h \in H_j} \hat{p}_{hj} \qquad (6)$$

Since $\hat{Y}_j$ represents the average per capita income of the households within a census tract $j$, we apply a similar logic to compute the model's mean per capita income prediction of the images within a census tract. For each census tract, we develop a grid of images of the same size as the ones used in the training set (50x50m + 200x200m). In other words, we

make predictions for the entire area of the census tract. Specifically, we only use the images whose centroid is located in the tract to create the grid. Therefore, some images might cover areas belonging to multiple tracts.

Fig 7 shows an example of a census tract over which we develop the grid to make the predictions. The $\hat{P}_j$ average prediction of the tract $j$ is made by averaging the income of the images from the constructed grid.

## 5.4 Model architecture and configuration

We employ the EfficientNetV2 architecture for our neural network [5]. These state-of-the-art models can achieve a high performance with a relatively low number of parameters. We use the Nesterov-accelerated Adaptive Moment Estimation as optimizer [34]. We trained using a Ge-Force RTX 2060 Super GPU and each training run required about 52 hours. All the scripts were written in Python using the Tensorflow-Keras library.

Several hyperparameter combinations were tested to select the most performant model. The tested hyperparameters were the size of the neural network, the learning rate, the size (in meters) of the image, the number of bands of the image (3, with Red-Green-Blue bands, or 4, which also includes the Near-infrared band) and the years used to train the model. All models were trained for 100 epochs due to hardware constraints. We selected the model iteration with the lower MSE in the test set at a census tract level, and further trained it with an early stopping of 10 epochs of no improvement in the test set, reaching 145 epochs.

## 5.5 Predicting income at grid level

Once the model is trained, we can obtain an income estimate from a single satellite image, without the need for any additional information. Therefore, by applying the neural network over a grid of satellite images of Buenos Aires in a given year, we can generate a highly disaggregated map of the income spatial distribution.

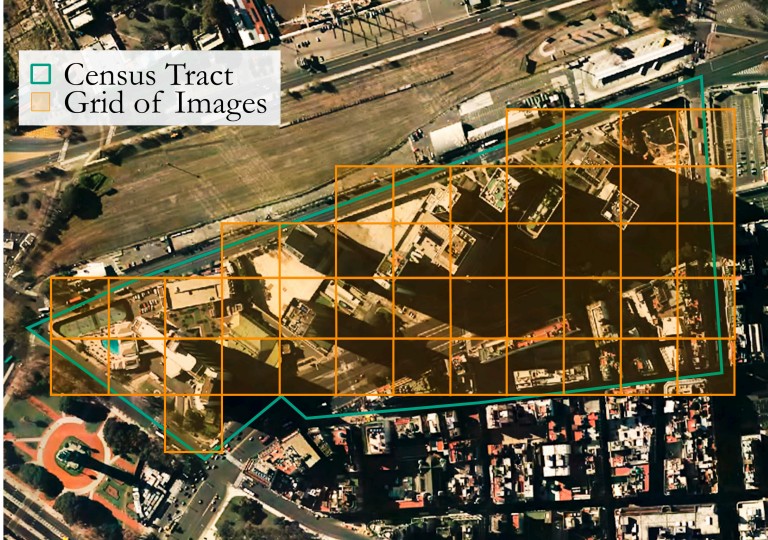

**Fig 7**. **Construction of a grid of 128x128px (50x50m) images to calculate the average income prediction in a census tract.** This figure illustrates the procedure used for model evaluation. A census tract (outlined in green) is systematically covered by a grid of 50x50 meter cells (orange). The model makes an income prediction for each cell, and the average of all these cell-level predictions is then calculated. This average represents the model's overall income estimate for the tract and is compared against the tract's known "ground truth" income to measure performance. The images shown are aerial imagery (from *Instituto Geográfico Nacional de la República Argentina*, 2025) for illustrative purposes as the underlying satellite data has licensing restrictions.

To make these predictions, we first need to create the corresponding grid for Buenos Aires, where each cell will be an image that the model evaluates. Once the grid is created, we make the predictions for every available image, both the ones belonging to the training set and the test set, from 2013, 2018, and 2022.

This mapping allows us to disaggregate the income indicator at a census sub-tract level. This is possible because the image selection for each tract is random. Since CNNs are robust to non-systematic errors in the input data [35], we expect that the model will overlook data incongruent with the general pattern in the training process. Therefore, our model can be applied in every census tract, producing consistent estimations. In addition, we can generate a time series of the income estimates for each cell of the grid previously created, since we can use the same model to analyze images from different points in time. In consequence, the trained model will enable us to develop maps of the indicator throughout time. As long as there are no widespread changes in the detected features—for example, what one would expect to see when comparing images of today and 50 years ago—, the estimates would remain consistent.

It's important to note that this approach can only detect changes in dwelling quality, urbanization, vegetation levels, and other similar factors. However, it cannot identify changes in income resulting from economic circumstances, such as short-term economic fluctuations, as the model does not have access to that type of information. Nonetheless, a tool like this is incredibly valuable for identifying both impoverished and affluent urban areas, and its evolution over time.

## 6 Results

As mentioned in previous sections, several hyperparameter combinations were tested to select the most performant model. The tested hyperparameters were the size of the neural network, the learning rate, the size (in meters) of the image, and the number of bands of the image (3, with Red-Green-Blue bands, or 4, which also includes the Near-infrared band), and the years used to train the model.

Table 3 shows the performance of all the tested combinations evaluated over the test set. The model that achieved the lower MSE (Eq 5) over the test set is an EfficientNetV2-S network with a learning rate of 0.0001, trained with images from both 2013, 2018, and 2022, using RGB and NIR bands and stacking images of 50$x$50 meters with 200$x$200 meters. Besides using the MSE for benchmarking each model, we have also assessed them qualitatively, by comparing the spatial correlation of income estimates in each case. A correct model should not only produce accurate predictions in terms of our benchmarking metric but also be in line with the general knowledge of how income is spatially distributed. Therefore, one should expect that, on average, adjacent cells have similar incomes. The combination of images of 50$x$50 meters with 200$x$200 meters achieves both the lowest loss and the most spatially correlated estimates.

The trained neural network enables the generation of very high-resolution predictions of per capita income. The model produces income estimates on a grid of 50 $\times$ 50 meters. In Argentina, no per capita income data is available at this level

**Table 3**. Model selection: configurations and mean squared error on the test set.

| Model Arquitecture | Image years | Image Size (meters) | Bands | MSE |
|---|---|---|---|---|
| EfficientNetV2-S | 2013 | 100x100 | RGB+NIR | 0.1152 |
| EfficientNetV2-M | 2013 | 100x100 | RGB+NIR | 0.1143 |
| EfficientNetV2-L | 2013 | 100x100 | RGB+NIR | 0.1123 |
| EfficientNetV2-S | 2013 & 2018 | 100x100 | RGB+NIR | 0.1032 |
| EfficientNetV2-S | 2013, 2018 & 2022 | 100x100 | RGB+NIR | 0.1051 |
| EfficientNetV2-S | 2013, 2018 & 2022 | 100x100 | RGB | 0.1063 |
| EfficientNetV2-S | 2013, 2018 & 2022 | 50x50 | RGB+NIR | 0.1157 |
| EfficientNetV2-S | 2013, 2018 & 2022 | 200x200 | RGB+NIR | 0.1132 |
| EfficientNetV2-S | 2013, 2018 & 2022 | 50x50+100x100 | RGB+NIR | 0.1078 |
| EfficientNetV2-S | 2013, 2018 & 2022 | 50x50+200x200 | RGB+NIR | **0.0923** |
| EfficientNetV2-S | 2013, 2018 & 2022 | 100x100+200x200 | RGB+NIR | 0.1025 |

of resolution. Therefore, to evaluate the model's performance, it is necessary to consider different levels of aggregation in order to verify whether the model generates predictions consistent with the available income data for the Buenos Aires Metropolitan Area. Additionally, it is important to visually verify that the model's predictions correspond with what is observed in the images. In Sect 6.1, we compare the model's predictions with the per capita income of each census tract. In Sect 6.2, we analyze the model's predictions within the census tracts qualitatively, given the lack of data at this resolution. In Sect 6.3, to verify if estimates are consistent across different levels of aggregation, we compare (i) predicted income aggregated for the whole city of Buenos Aires and (ii) predicted income aggregated at the municipal/communal level, with data from the Permanent Household Survey and small area income estimates from the 2010 census. Finally, in Sect 6.4 we assess the validity of the model's predictions over time by comparing the results for 2013, 2018, and 2022.

## 6.1 Quantitative evaluation of model results at census tract level

Formally, our predicted variable is the standardized logarithm of the per capita average income of the cell, and we show the deciles of the 2010 income distribution in the figures for easier interpretation. Fig 8 shows, on the left panel, the predictions for the grid of satellite images from 2013. In the right panel, it shows the original data—the labels of the test and training sets derived by applying small-area estimation techniques to the census and survey data. These figures illustrate the primary benefit of this methodology: the ability to enhance the resolution of income estimation by leveraging the unstructured data available from satellite images.

The proposed model has a great predictive performance, reaching an $R^2 = 0.878$ over the test set. In other words, the model manages to capture more than 87% of the variability of the spatial income on a set of images that the model has never seen. This high predictive capability becomes evident in Fig 9, which compares, for each census tract, the logarithm of the income of each tract (derived from small area techniques) compared with the average income prediction from the neural network. Both variables are standardized in order to get a mean of 0 and a standard deviation of 1. The model predicts with few errors for the lower-income tracts. When dealing with low and middle-income levels, the model depicts a

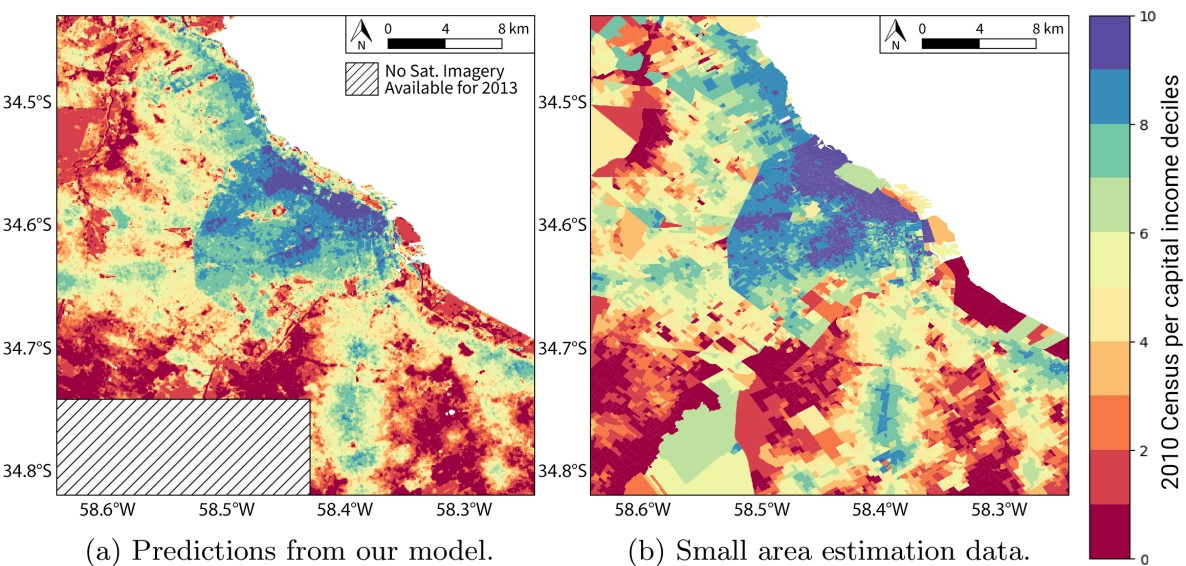

(a) Predictions from our model.    (b) Small area estimation data.

**Fig 8. Spatial per capita income estimates of Buenos Aires.** This figure highlights the primary contribution of the study by comparing the model's output with the input data. The map on the left shows the per capita income for Buenos Aires in 2013, as predicted by the model on a high-resolution 50x50 meter grid. The map on the right shows the lower-resolution 2010 "ground truth" income data at the census tract level, which was used for training. The comparison visually demonstrates the model's ability to create a significantly more granular and detailed income map.

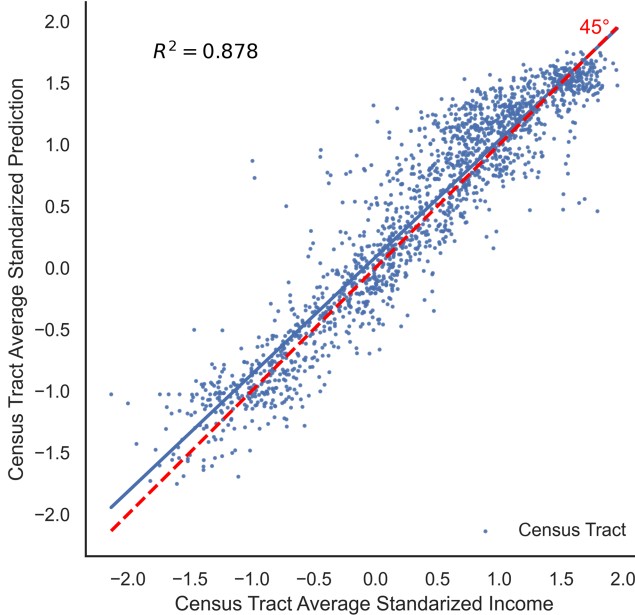

**Fig 9**. **Comparing predicted and observed income by census tract in the test set.** This scatter plot provides a quantitative evaluation of the model's accuracy on the test set. Each point represents a census tract, plotting the model's average predicted income (y-axis) against the "observed" income from small area estimations (x-axis), with both values being standardized logarithms. The strong linear relationship, with a coefficient of determination ($R^2$) of 0.878, demonstrates the model's high predictive power at the census tract level.

greater variability in its predictions. This is reasonable since low-income households have fewer opportunities to choose the house type. Thus, the observed variability in the satellite images is lower.

### 6.2 Qualitative evaluation of model results at grid level

Given the lack of available data at a census sub-tract level to assess the precision of the grid estimations, in this section we qualitatively assess our results. We present a series of case studies in Fig 10, to highlight complex situations that might challenge the model's predictive capacity, such as high-income gated communities next to informal settlements. All the cases come from the test set, to ensure that the model has not previously seen these images. Nonetheless, we argue that the results we obtain can also improve those of the training set, given the original data is composed of (a) estimations, not observed data, and (b) census tracts, and not individual households.

**Census tracts with high-income and geometric variability.** It is interesting that, in the presented cases, informal settlements, such as the Barrio 31 (in Retiro) and Barrio 21-24 (in Nueva Pompeya), are correctly identified by the model, delimiting an area where clearly there are poor houses in inhabited areas. In addition, the areas with buildings or houses built with better construction quality are classified as middle or high-income according to the census data.

**Emerging neighborhoods.** At the same time, as we can see in the case of Bellavista, some areas with emerging urbanization—such as the neighborhood with pools under construction shown in the center of the image—are classified as high-income, as opposed to the census data, which classify them as low-income. This can happen due to the fact that the census tract comprises many observations from one relatively poor section and just a few higher-income households. The model seems to correctly capture the "real" distribution of income, being able to detect the characteristics of high-income houses seen in more uniform census tracts.

**Census tracts with high income but low geometric variability.** Finally, the bottom panel depicts a case in which the same census tract of Tigre includes part of a gated community, with higher-income households, and part of an informal

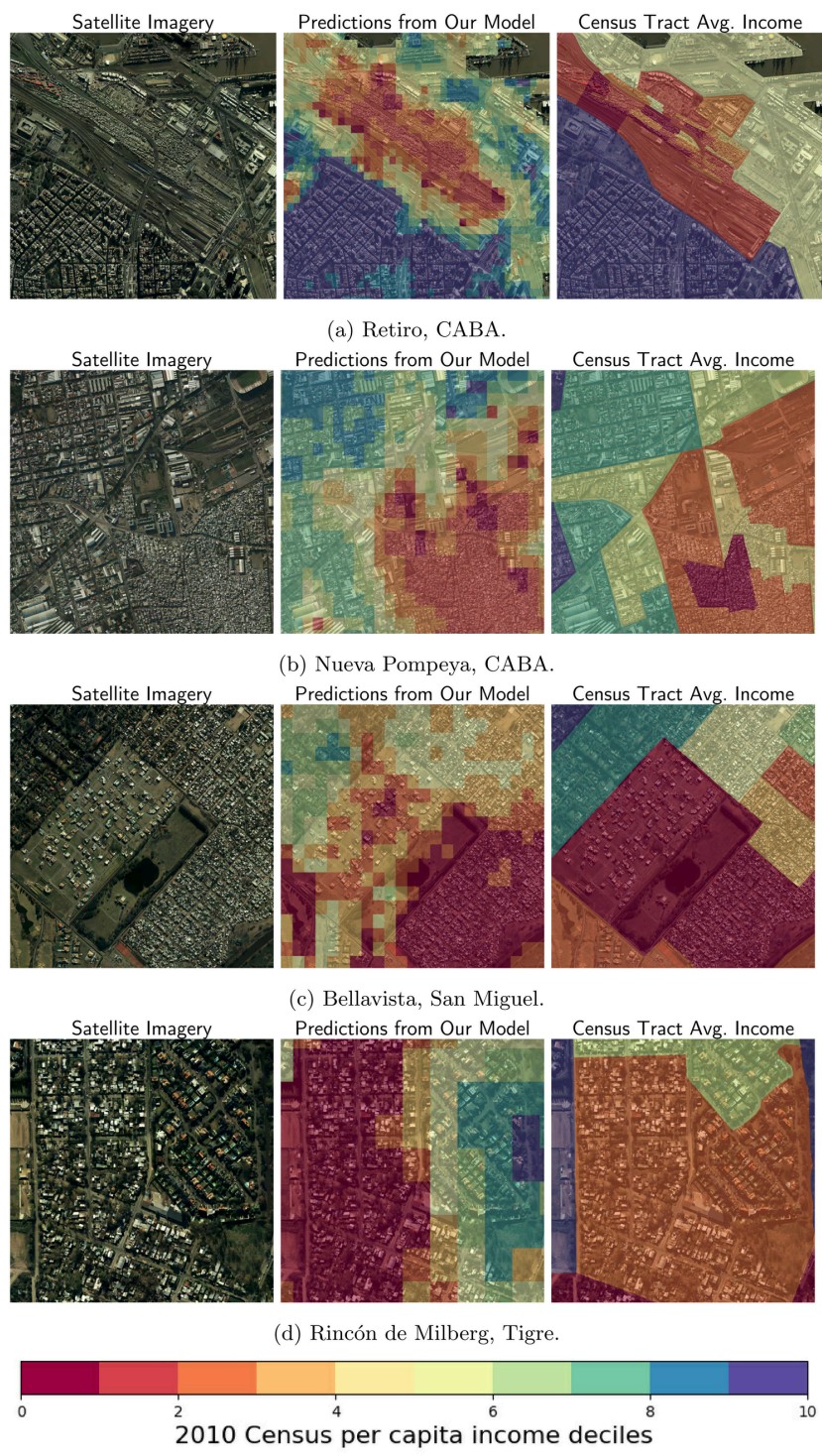

(a) Retiro, CABA.

(b) Nueva Pompeya, CABA.

(c) Bellavista, San Miguel.

(d) Rincón de Milberg, Tigre.

2010 Census per capita income deciles

**Fig 10**. **Case studies taken from the test set.** This figure presents several case studies from the test set to qualitatively evaluate the model's performance in complex urban environments. Each case consists of a satellite image (left) and the corresponding high-resolution predicted income map (right). The examples showcase the model's ability to accurately differentiate income levels, such as identifying low-income informal settlements adjacent to formal housing and correctly mapping income disparities, even within a single, heterogeneous census tract. The images shown are aerial imagery (from *Instituto Geográfico Nacional de la República Argentina*, taken between 2013 and 2022) for illustrative purposes as the underlying satellite data has licensing restrictions.

settlement with low-income households. On average, the census tract presents a middle-high income, since that might be the average income of all the households in such tract. As it is portrayed in the figure, our model can better capture the income spatial distribution, since it allocates the middle-high incomes in the gated community cells and the low incomes in the informal settlements cells. In a case like this one, if we compare each image with the average census tract income, the error will be bigger than the one calculated by comparing the predicted average income with the average census tract income.

## 6.3 Validation via prediction of economic indicators for 2010

As we previously mentioned, the main objective of the proposed model is to capture the income spatial distribution with a high spatial resolution. However, it is important to assess if, after adding the grid predictions to higher geographical levels, the model estimates precisely some relevant economic indicators. Thus, we compare the predictions of the model with indicators derived from survey data and the small area estimates from the 2010 Census.

In Table 4 we compare descriptive statistics of the income distribution of (1) the 5348 households surveyed in the 2010 second semester in Buenos Aires city, (2) the census income prediction for 13,491 census tracts from Buenos Aires, that is, the small area estimation for each tract (see Eq 3) and (3) the income prediction based on satellite images for the grid of images from Buenos Aires, a grid of 539,809 cells.

In cases (2) and (3), where we observe areas rather than households, we calculate the indicators by weighing each area according to the number of households that live there. In the census, this information is available for each census tract. For the grid level predictions, we used the population estimates from Gridded Population of the World project [2], which estimates the population in a grid with a 30-arcsecond resolution (around 1 kilometer). This is done to avoid focusing too much on areas with low or zero population density, such as Buenos Aires outskirts. If we do not do this, the average income estimation would be excessively low, underestimating the "real" average income.

**Table 4**. **Performance on predicting economic indicators.**

| | (1) Survey Data | (2) Census Estimates | (3) Our Model |
|---|---|---|---|
| Unit of analysis | Households | Census Tracts | Cells |
| N° of Observations | 5,348 | 13,491 | 539,809 |
| Year | 2010 | 2010 | 2013 |
| Spatial resolution | Citywide | 300x300m[a] | 50x50m |
| Potential Data frequency | Quarterly | Decennially | Monthly[b] |
| Indicators: | | | |
| *Mean* | 820.94 | 691.84 | 649.37 |
| *Median* | 611.45 | 567.16 | 541.75 |
| *90/10 Ratio* | 8.05 | 3.59 | 3.78 |
| *50/10 Ratio* | 3.01 | 1.62 | 1.74 |
| *Gini* | 0.43 | 0.31 | 0.28 |
| *Theil* | 0.32 | 0.15 | 0.12 |
| *Atkinson (1)* | 0.28 | 0.15 | 0.11 |
| *Atkinson (2)* | 0.50 | 0.30 | 0.20 |
| *FGT(0)* | 32.17 | 30.30 | 37.01 |
| *FGT(1)* | 12.92 | 4.70 | 7.58 |
| *FGT(2)* | 7.00 | 1.02 | 1.99 |

[a] The spatial resolution of census estimates are the census tracts, which, on average, have 300x300m size on high population density areas, but there is a large variability between tracts sizes.
[b] The only limiting factor for creating time series with our model is the availability of the imagery. Given that Pleiades Neo has a twice-a-day revisiting factor, one could theoretically produce estimates at this time resolution. Of course, this depends on weather conditions and particular needs of data, so a monthly frequency seems more reasonable, although in certain cases even daily data could be produced.

We compare the following indicators of income distribution (for a detailed discussion on each indicator, refer to [36]): mean and median incomes; the 90/10 and 50/10 Ratios; the Gini coefficient; the Theil index; the Atkinson-1 and -2 indices; and the Foster-Greer-Thorbecke (FGT) poverty indicators (FGT(0), FGT(1), and FGT(2)). Mean and median incomes provide a sense of the typical income level. The mean gives the average, while the median represents the income of the middle household, which is less affected by the high skewness of the income distribution—as there are many households with low income but a few with very-high income—, and thus better reflects the conditions of the 'central' household. The 90/10 and 50/10 Ratios specifically highlight the spread of the income distribution by comparing different segments. The 90/10 ratio shows the disparity between the income of the household at the top and the bottom 10% of the income, while the 50/10 ratio illuminates the gap between the median household's income and the bottom 10% income. These ratios are useful for understanding how different parts of the income spectrum relate to each other, offering more granularity than a single summary statistic. Nevertheless, to obtain a more nuanced vision of the income distribution, economists often use indicators that depend not only on the income of a particular household but on every single one. The Gini coefficient provides a comprehensive single number that summarizes overall relative income inequality across the entire population. While widely used for its conciseness in comparing inequality levels, it's important to note that different income distributions can yield the same Gini coefficient, which is why it's complemented by other measures that reveal more about the shape of the distribution. The Theil index is another measure of relative inequality. Its unique strength lies in its decomposability, meaning it can be broken down to show how much inequality is due to differences between pre-defined groups (e.g., urban vs. rural, different education levels) versus inequality within those groups; while we don't perform such decomposition in this specific table, its inclusion represents a standard tool for deeper inequality analysis [37]. On the same line, the Atkinson indices (e.g., Atkinson (1) and Atkinson (2) in our table) measure inequality but uniquely incorporate a concept of 'inequality aversion.' This means they explicitly reflect how much a society might care about incomes at the lower end of the distribution, with different parameters of the Atkinson index (represented by the numbers in parentheses) allowing for varying degrees of emphasis on the poorest, making it a flexible tool for welfare-based inequality assessment [38]. Finally, we focus on absolute poverty indicators. The Foster-Greer-Thorbecke (FGT) family of poverty indicators offers a versatile framework for measuring different dimensions of poverty, calculated with the national poverty line from the *Socio-Economic Database for Latin America and the Caribbean* [39]: FGT(0), the poverty headcount ratio, is the most straightforward measure, indicating the percentage of the population whose income falls below a defined poverty line; FGT(1), the poverty gap index, goes beyond mere incidence to measure the depth of poverty, reflecting the average amount by which poor households' incomes fall short of the poverty line; and FGT(2), the squared poverty gap index, measures the severity of poverty, giving greater weight to those who are furthest below the poverty line and thus being particularly sensitive to the plight of the poorest among the poor [40].

We can draw many conclusions from Table 4. In the first place, the measures of the central trend derived from our model depict an underestimation regarding the survey data. This is particularly true for the mean income, which goes from an average household per capita income of $ARS_{2010}$ 820.94 to a predicted average income of $ARS_{2010}$ 649.37. It is possible to explain this if we consider that the heterogeneity of houses, observed by the model, can be lower than the heterogeneity of income. For instance, this would happen if the house income elasticity of demand is lower than 1, something seen in, for example, the United States [41] and Mexico [42]. Moreover, when the model's predictions seek to minimize the MSE—such as the linear regression in the Eq 1—, the distribution variability is significantly compressed, reducing the inequality indicators. In an asymmetric distribution, such as income, the average is also expected to decrease in the face of this variability drop, by reducing the influence of extreme incomes over this indicator. In Fig 9, both consequences combined can be seen clearly if we consider that, while the census tracts income along with the census data of most observations are between –2 and 2 standard deviations of the average, most predictions are between –1,5 and 1,5.

For both reasons, inequality measures are also expected to be underestimated, as it is highlighted when comparing the indicators for models (2) and (3) regarding the data observed in the survey, column (1). Poverty indicators have the same problem: with the processing of both models, the poverty measures cannot be estimated correctly, particularly those more

sensitive to the distribution of the left tail, such as the FGT(2). It is important to emphasize that this essential underestimation of the inequality and poverty indicators is explained when going from the column (1) data to the column (2) data, that is, when making small area estimations of the census tracts. In contrast, CNN makes predictions very similar to the data with which it was trained (column (2)). In fact, the neural network might be able to capture non-incorporated variations in the small area model, explaining why the poverty and inequality measures are more similar if columns (3) and (1) are compared than if columns (2) and (1) are compared, the only exception being the FGT(0). This is especially remarkable if we consider that, in order to make the CNN predictions, the data used for the training process were the ones from column (2). In other words, the created model manages to capture variations in the spatial input that were not incorporated in the original data.

When the income at a county/municipality level is considered, the conclusions are similar. Fig 11 compares, for each county in the city of Buenos Aires, different economic indicators computed with the census data, with the same indicators computed with the income predicted by the CNN for each grid cell. In other words, we are comparing the predictions, aggregated at the county level, of the models depicted in columns (2) and (3). We can extract three main conclusions. Firstly, at a municipal level, the satellite image estimations systematically underestimate the census mean and median income. This explains why the model also underestimates inequality and poverty, as all the predicted incomes are generally lower than the real ones. Secondly, the model correctly captures the trend of every indicator at a municipal level, since, for instance, municipalities that face greater inequality according to the census data also deal with a higher estimated inequality according to the satellite images. Thirdly, it is worth highlighting that the estimations of the central trend measures are by far the most precise, creating an indicator that, although it underestimates the income, it captures with a high level of accuracy the income ordering of the municipalities. In other words, the model appropriately captures (at a municipal level) the income spatial distribution, but not households' per capita income.

### 6.4 Income predictions through time

By applying the same trained model from previous sections, it is possible to create an income prediction grid for the images from 2018 and 2022. To do so, we simply have to make the predictions for each image of the generated grid of Buenos Aires city, in each of the corresponding years.

Fig 12 shows the evolution over time of the per capita income, expressed in income deciles from 2010, in the three years analyzed. To the left, we replicate Fig 8, so it is easier to compare the images from previous years. In general, the model predicts a very similar spatial income distribution for the three years, which highlights the consistency of the model when predicting with images from different years. On the other hand, for 2018 and particularly 2022, a lower spatial correlation between the near cells is shown. While the images from 2013 show more consistent clusters of cells that belong to the same decile, in the images from 2022, the predictions for a specific area are more varied. In Sect 8, we briefly discuss why this might be happening.

By applying the same trained model from previous sections, it is possible to create an income prediction grid for the images from 2018 and 2022. Fig 12 shows the evolution over time of the per capita income, expressed in income deciles from 2010, in the three years analyzed. To the left, we replicate Fig 8, so it is easier to compare the images from previous years. In general, the model predicts a very similar spatial income distribution for the three years, which highlights the consistency of the model when predicting with images from different years. On the other hand, for 2018 and particularly 2022, a lower spatial correlation between the near cells is shown. While the images from 2013 show more consistent clusters of cells that belong to the same decile, in the images from 2022, the predictions for a specific area are more varied. In Sect 8, we briefly discuss that this might be happening.

At the grid level, the model seems to appropriately predict the evolution of urban infrastructure over time. As an example, one of the cases presented in Sect 6.2, corresponding to the locality of Bella Vista, San Miguel, is offered. Fig 13 shows the evolution of the predicted income between 2013 and 2018 for an image corresponding to the test set, meaning

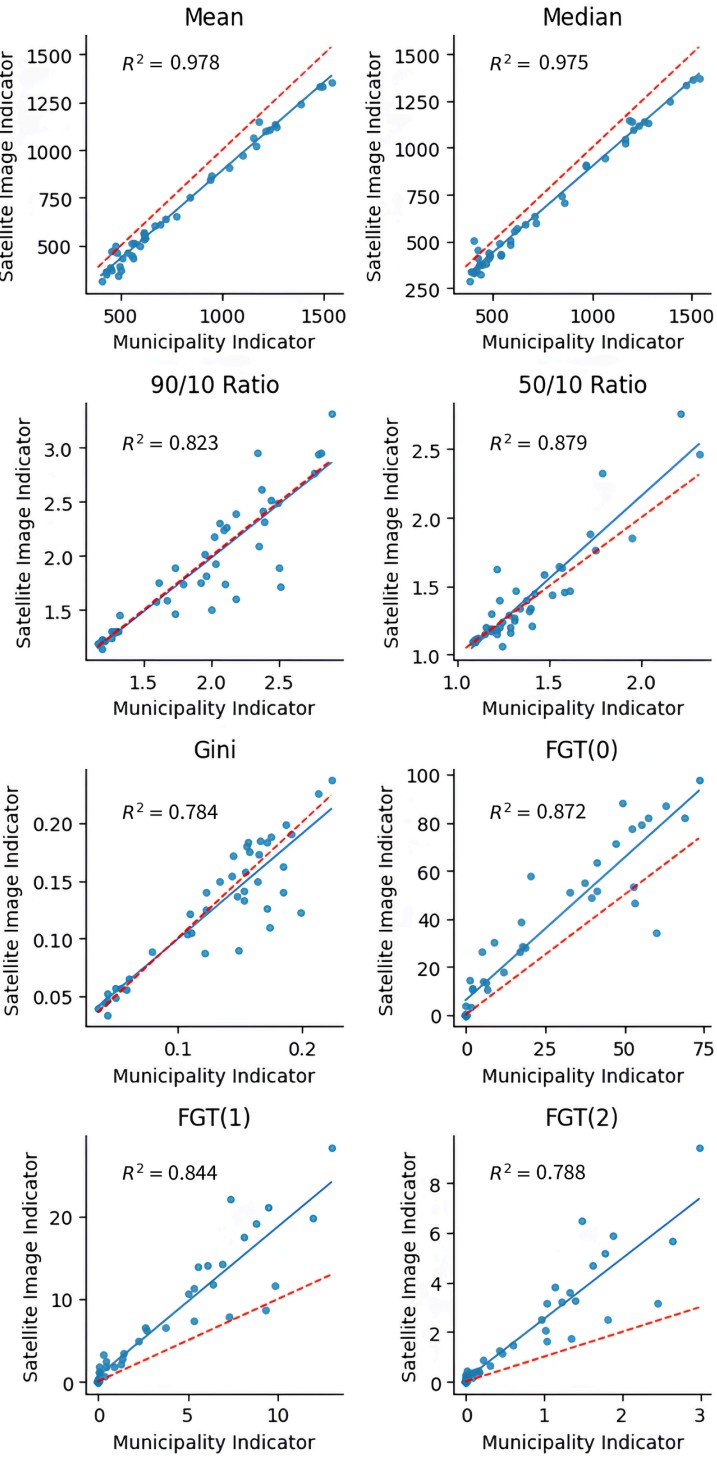

**Fig 11. Comparing indicators at county level.** Note: each dot represents a county. This figure contains a series of scatter plots comparing various economic indicators (e.g., mean income, median income, Gini coefficient) at the aggregated county (municipality) level. For each plot, the x-axis represents the indicator calculated from the 2010 census-based estimates, while the y-axis shows the same indicator calculated from the model's 2013 high-resolution predictions. The plots demonstrate that the model's predictions strongly correlate with the census-based data, correctly capturing the relative ranking and trends across municipalities.

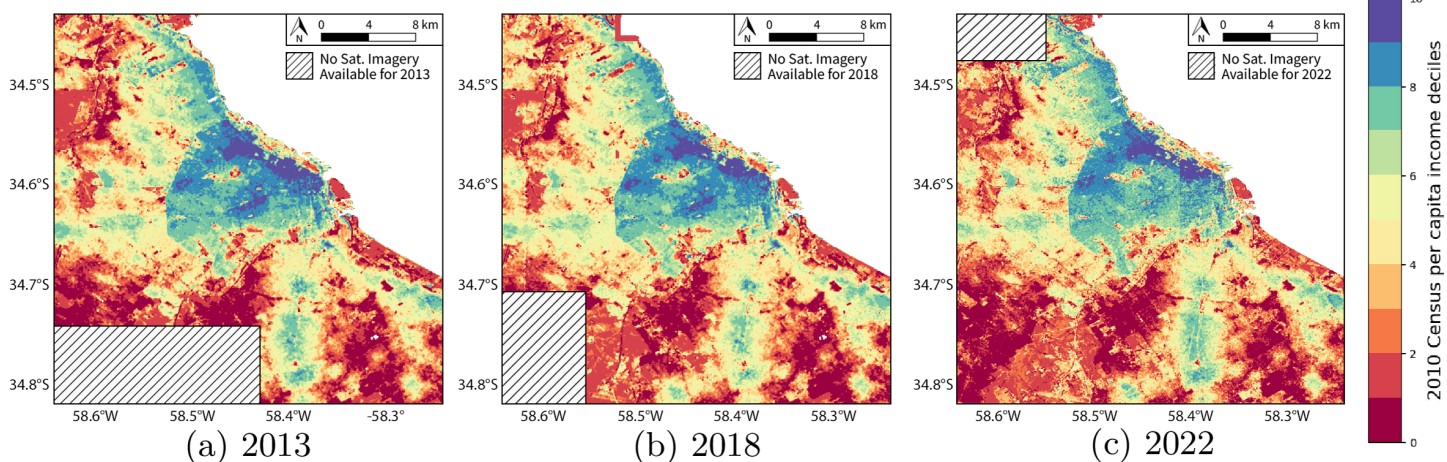

**Fig 12**. **Per capita income predictions over time.** This figure presents three predicted per capita income maps for the Buenos Aires Metropolitan Area for the years 2013, 2018, and 2022. All three maps were generated using the same trained model, applied to satellite imagery from each respective year. The results demonstrate the model's ability to produce temporally consistent spatial income distributions, allowing for the analysis of urban socioeconomic patterns over time. The color scale represents income deciles based on the 2010 Census.

it was not previously seen by the model. As can be observed on the left side of the images, there is an area of incipient urbanization in 2013 that is already well-developed by 2018. In 2013, many of the houses are under construction, many without roofs, so the model predicts medium and low incomes—areas with unfinished and/or abandoned buildings are typically inhabited by lower-income households. By 2018, urbanization is more defined, with large houses, accompanied by pools and large gardens, typically inhabited by higher-income households. The model successfully captures this dynamic, while the surrounding areas that did not experience changes remain relatively consistent across time.

In the previous sections, we estimated a series of economic indicators for the year 2010 for the hole Buenos Aires city. Here, we compute this same indicators for the income grids for 2018 and 2022. As expected, these results have their limitations when compared with the income observed in surveys data, since the current income can be affected by short-term variations and most of the observable characteristics of the satellite images are related to medium and long-term income variations.

Fig 14 shows the percentage variation of different income indicators over time, using our model's satellite image predictions and the survey data from the corresponding semester. Although the indicator's growth rate is not perfectly captured by the model—in every case, the model predictions are a few point behind the survey data—, it is worth emphasizing that for every indicator, the trend is captured by satellite images. Only in a few cases, such as the Ratio 50/10 and the Gini coefficient for 2018, the model incorrectly predicts the temporary evolution of the indicator. In addition, it is important to highlight that the indicators predicted by the model systematically underestimate its evolution through time. This goes hand in hand with the interpretation that the model does not capture the common income of people, but rather their long-term or permanent income. We discuss this in detail in Sect 7.

## 7 Conceptualizing the model's measurement of income

In this paper, we presented a model that generates highly detailed spatial income estimates for Buenos Aires using high-resolution satellite imagery. The model identifies observable features from the images—such as building materials, urbanization patterns, vegetation, and road networks—and use them as proxies for estimating the per capita income of the

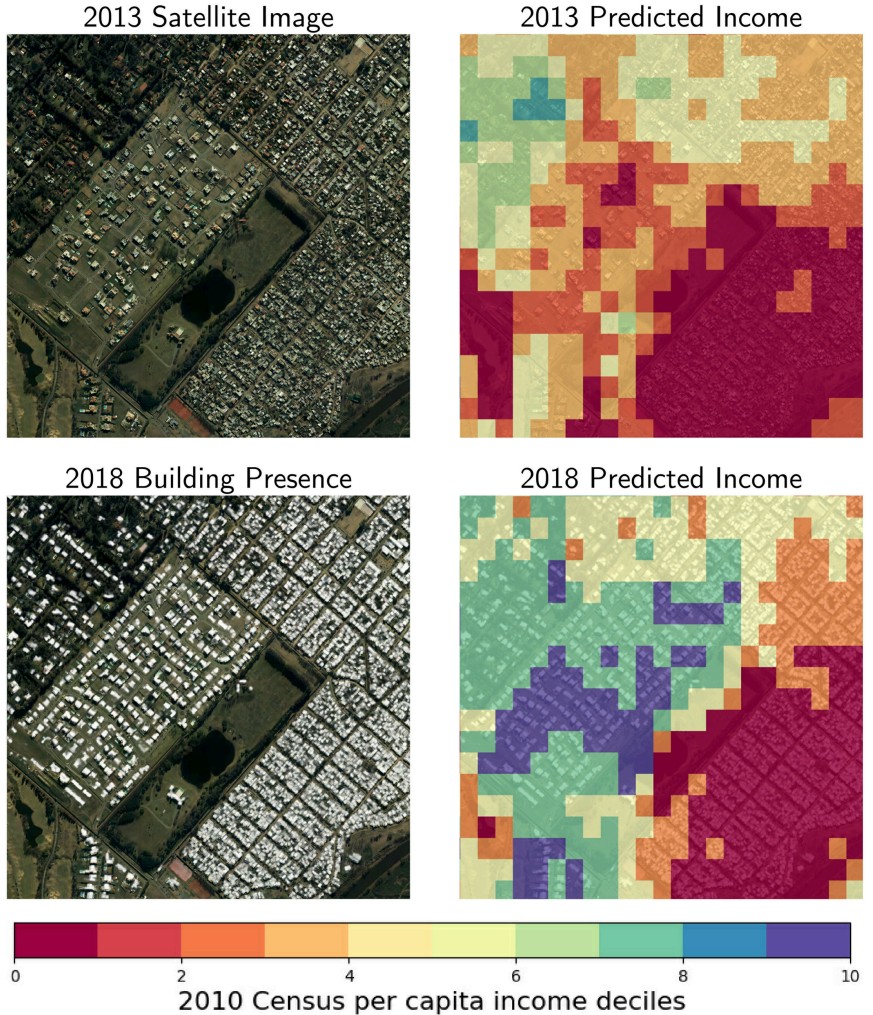

**Fig 13**. **Detecting development over time in Bella Vista, San Miguel.** This case study from the test set shows the model tracking socioeconomic change from urban development in Bellavista, Buenos Aires. The top row displays a 2013 aerial image and the corresponding income prediction. The bottom row visualizes new construction by 2018, and the model correctly identifies the developed area as high-income. The heatmap color scale corresponds to the 2010 Census per capita income deciles. The 2013 aerial photo (from *Instituto Geográfico Nacional de la República Argentina*, 2025) and 2018 building layer (from the *Open Buildings Dataset* [43]) are shown for illustration, as the original satellite imagery used for prediction is not licensed for publication.

average household living in the area of such image. The challenge lies in what these observable features represent in economic terms.

The most immediate interpretation of the model's output would be that it measures current income, i.e., the income a household or individual earns at a given moment, as typically captured by household surveys (Beccaria y Gluzmann, 2013). However, this is not a straightforward equivalence. Current income, often measured through surveys, is inherently volatile. It captures short-term fluctuations and can vary significantly due to temporary shocks, seasonal variations, or even job transitions. For example, a professional who is temporarily unemployed may show low income in a given month, but this does not reflect their true, long-term economic well-being. Likewise, households may experience periods

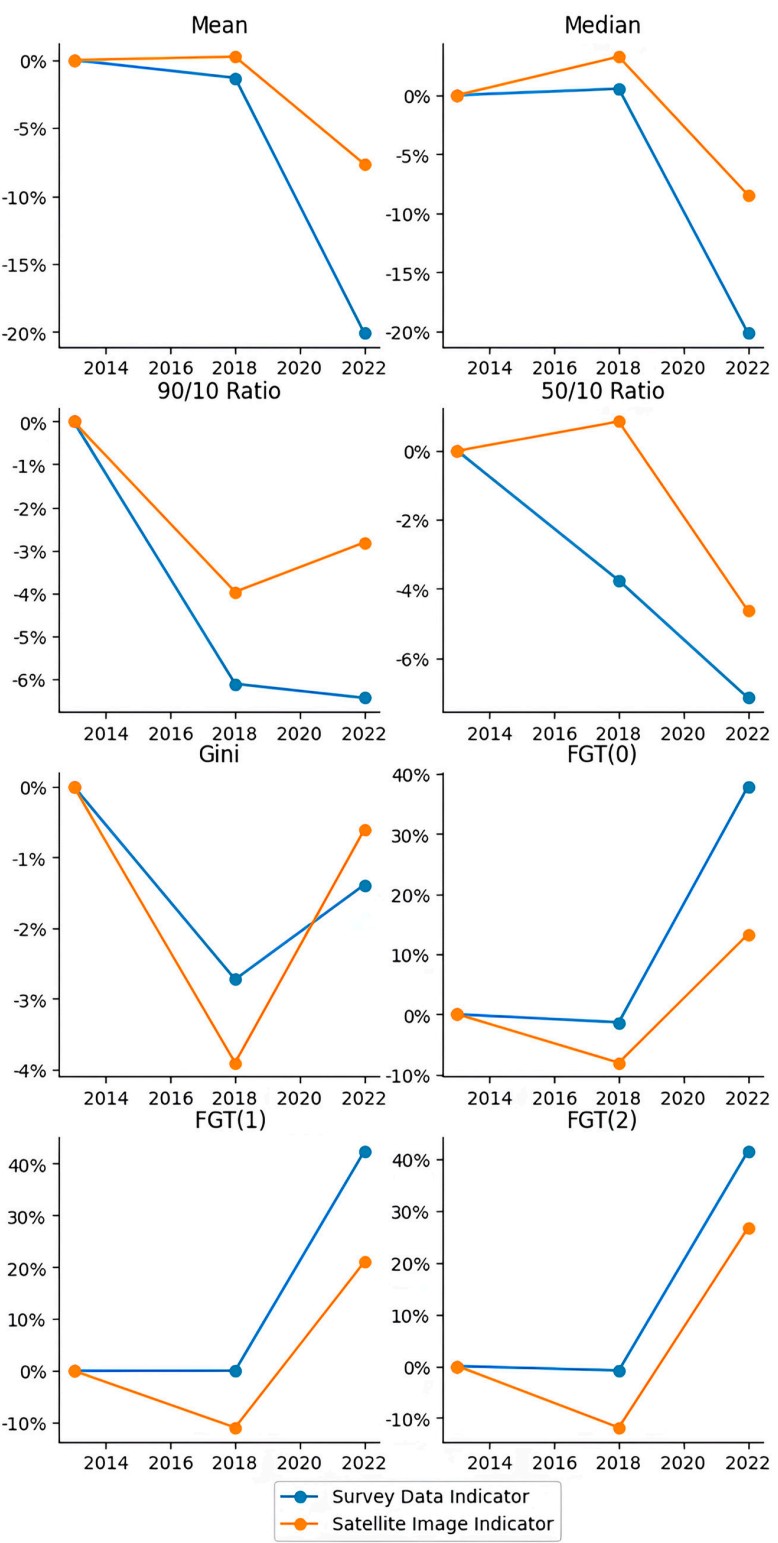

**Fig 14**. **Growth rates of income indicators over time.** This line chart compares the temporal change of various economic indicators as measured by two sources: official household survey data and the model's satellite-based predictions. For each indicator (e.g., mean income, Gini, poverty rates), the chart shows the percentage change between years. The results indicate that the model's predictions successfully capture the direction of change (the trend) for most indicators, suggesting its utility for tracking socioeconomic evolution.

of income volatility due to factors like temporary job loss, illness, or cyclical economic conditions. This volatility is one reason why household surveys often show individuals with very high levels of education living below the poverty line (Gasparini et al., 2013).

The model, which analyzes structural features of urban areas, is more likely to capture these long-term economic conditions rather than the transient income changes that surveys typically capture. Drawing from the Permanent Income Hypothesis (Friedman, 1957; Hall 1978), which posits that individuals base their consumption decisions on their expected long-term income rather than short-term fluctuations, we argue that the model's predictions likely represent permanent income rather than current income. Permanent income reflects a more stable view of economic well-being. It accounts for the average income level that a household can expect over time, disregarding temporary changes that may not affect the household's material conditions in the long run. The observable features in satellite images—such as housing quality, the presence of infrastructure, and the extent of urban development—are more indicative of a household's permanent income rather than its current or transient income. This is a significant distinction because permanent income is a more stable and structural measure of economic welfare, which can be used for long-term policy planning, targeting social programs, and understanding the socio-economic conditions of urban populations.

One limitation of this approach is that it cannot capture short-term fluctuations in income. Households that experience temporary changes in their income, such as a temporary job loss or seasonal variations, will not be accurately represented by the model, as the satellite imagery and the model's architecture focus on structural, long-term characteristics. While this is a strength in terms of providing a more consistent measure of welfare, it also limits the model's ability to assess more transient changes in well-being, which are also crucial for certain types of economic analysis and policy interventions.

## 8 Generalizing the model to other cities

The methodology described can be replicated straightforwardly when satellite imagery, along with census or survey microdata, is available. Even more, if spatially disaggregated income data exists, the model training can directly use that data, bypassing some of the intermediate steps detailed in Sect 4.1. However, when the quality or quantity of available information is limited, or the computational resources for training such a model are not available, adapting the model trained for Buenos Aires to a new urban context becomes essential.

A critical step in transferring the model to another city involves domain-specific fine-tuning, a widely accepted practice in transfer learning [44]. This process involves leveraging the feature representations learned by the pre-trained model on the extensive Buenos Aires dataset and adapting them to the unique characteristics of the target city. Typically, the earlier convolutional layers, which identify basic visual elements such as edges, textures, and shapes, are either frozen or trained with a very low learning rate. In contrast, the later, more specialized layers, particularly the fully connected layers responsible for classification or regression, are retrained using image-income data pairs from the new city.

This fine-tuning is essential because the relationship between visual features and income levels can vary significantly across cities. For example, architectural styles, roofing materials, vegetation, and urban layouts that correlate with specific income levels in Buenos Aires may have different associations in a city shaped by distinct cultural, climatic, or historical factors. Moreover, the absolute income levels linked to visually similar urban features can differ substantially across cities, influenced by differences in economic prosperity, cost of living, and income distribution. For instance, a multi-story residential building with certain visual characteristics may represent an upper-middle-income property in Buenos Aires but correspond to a lower-middle-income or even low-income dwelling in a city with a higher per capita GDP.

A similar procedure can be applied for generating income estimates in cities where spatially disaggregated income data is unavailable. In such cases, data from cities with comparable income levels and spatial characteristics can be used to fine-tune the Buenos Aires model. With the updated parameters, and images from the cities lacking income data, the model can generate accurate income predictions for those areas.

In addition, incorporating local non-visual data sources can significantly enhance the model's accuracy. The final layers of a neural network, known as the "fully connected layers," are designed to integrate complex visual information and make the final income prediction. These layers can also incorporate numerical data from economic indicators that might help improve the estimation. For instance, adding population density, consumption metrics (e.g., electricity usage), or data on urban development and migration trends can help capture income fluctuations and socio-economic changes that satellite imagery alone may miss. *Khachiyan et al* [16] demonstrated this by adding indicators like average annual income to the final CNN layer, greatly improving their model performance. Similarly, *Rao and Molina* [45] employed a similar approach by integrating survey indicators into small area estimations.

There are two primary limitations that must be carefully considered when producing estimates using this methodology.

First, during the model training, it became evident that predictions were significantly influenced by contextual factors unrelated to actual income levels. For instance, when the model was trained exclusively on images from 2013 and then tested with images from 2013, 2018, and 2022, it consistently predicted lower incomes for the 2018 images. This outcome occurred because the 2018 images were captured during a drier month, featuring yellower vegetation typically associated with lower-income areas. Conversely, the model predicted higher incomes from the 2022 images due to shadows cast by buildings photographed in the evening, implying the presence of taller structures usually indicative of higher-income neighborhoods. To mitigate such idiosyncratic fluctuations, it is essential to incorporate a substantial and varied dataset of images from multiple time points, especially when generalizing this approach to other urban contexts.

Secondly, it is crucial to emphasize that the current model does not discern whether households actually exist within a given grid cell. Consequently, the model's predictions have limited utility in areas with sparse or no population, as such irrelevant data introduces noise into gradients and predictions. To address this shortcoming, implementing a two-stage modeling approach is advisable. The first stage would predict the presence of at least one household within the grid cell, while the second stage would subsequently estimate income levels only if the presence of housing is confirmed.

The primary potential application for datasets generated by models such as this lies in the measurement of localized development, offering valuable insights for urban planning and the design of targeted interventions aimed at improving infrastructure or services. However, it is crucial to emphasize that these models, at least in their current state of development and validation, should not be directly used for high-stakes individual-level decisions such as taxation or determining eligibility for social programs without extremely careful consideration of the ethical aspects involved [46]. Beyond these specific applications, broader ethical implications must be thoroughly addressed. Biases present in the original census or survey data, or in the small area estimation process itself, could be perpetuated or even amplified by the model, potentially leading to unfair resource allocation or misrepresentation of vulnerable communities. Privacy concerns, though somewhat mitigated by the use of aggregated census tract data for training, remain pertinent. There is also the risk of misuse, where such granular income data could be leveraged for discriminatory practices, such as redlining or targeted marketing that exploits socio-economic vulnerabilities. Therefore, the deployment of this technology must be accompanied by robust frameworks for ethical oversight, including transparency regarding model limitations, the incorporation of methods for model explainability to understand how predictions are made, ongoing bias audits, community engagement in areas where the data is applied, and a clear commitment to using these powerful tools to foster equitable development and well-being.

In summary, while the model developed for Buenos Aires is inherently localized and faces some key limitations, the strategies outlined here provide a clear pathway for applying and refining this high-resolution income estimation methodology in diverse urban settings worldwide. The goal of the presented model was to establish a robust framework for high-resolution income estimation using satellite imagery, rather than to provide the "best" possible estimates by incorporating all available local data for Buenos Aires. However, this foundational approach is designed to be adaptable and can be further enhanced when applied to new urban environments.

## 9 Accessing the generated datasets

The income maps developed for 2013, 2018 and 2022 are available here and in the supporting information (S1 and S2 Dataset). Since the predictions for each cell individually present some random variation, we recommend that the results are used averaging out the estimations for each area of interest (municipalities, neighborhoods, sections or census tracts) and not at an individual level. As we detailed throughout this work, the aggregated results, even in small areas such as census tracts, predict in a precise way the households' income. Likewise, inside the repository, it is possible to access and use the model's trained parameters to make predictions about different satellite images.

## 10 Conclusion

In this study, we present a methodology for developing a Machine Learning model capable of generating highly detailed spatial income estimates using satellite images. In addition to this method, we outline a strategy for constructing a database that can be utilized with a Convolutional Neural Network using georeferenced census microdata, even if no income data is collected at the census. The results demonstrate that the model achieves high precision in predicting per capita income, surpassing 20 times the spatial resolution of alternative methodologies found in the literature.

The application of this model allows us to generate a map of the per capita income of the Metropolitan Area of Buenos Aires for the years 2013, 2018, and 2022, improving the spatial resolution of the original census data. This provides a more precise vision of the income distribution in the studied region, at years where no data is available at such spatial level. It helps identify areas with low-income populations, which is valuable for both academic research and public policy.

The implications of this work are significant at various levels. The methods outlined here can be applied in numerous cities across countries with varying income levels to support the development and assessment of public policy interventions and social programs. The only two requirements are that census and survey microdata is available, which is true for many countries around the world, and high-resolution satellite imagery is available or purchased. Furthermore, if a model of this kind is trained over a wide range of cities across the world, it could help bridge the data gap in low-income regions by predicting the income of cities in countries without survey data, such as many Sub-Saharan African countries, only using the satellite imagery as data source. Finally, focusing on Buenos Aires, providing public access to a grid database with geographical income information, will enable diverse applications in various fields beyond economics.

The ability to generalize the model for other urban areas will help to close the data gap in low-income areas, including cities in countries with no survey data, such as most countries of Sub-Saharan Africa [35]. The methods presented here can be replicated to many cities in both high, middle and low income countries where census microdata is collected, in order to help the design and evaluation of public policy interventions and social social programs. Furthermore, focusing on Buenos Aires, making a grid database available for the general public provides access to income geographical information at an unpreceded level, enabling numerous applications in different disciplines besides economics.

The development of this methodology presents four key areas for future investigations with the aim of improving these models. In the first place, it is essential to increase the satellite imagery datasets beyond Buenos Aires. As satellite technology advances and the related costs decrease, this expansion will become more feasible in the near future. Secondly, adding images from many different years instead of only a few points in time will allow us to develop better time series predictions and, thus, create temporal maps for the evolution of the geographical distribution of income. In addition, the subsequent predictions could improve the quality of the estimation, since random differences in the image from a specific year could be filtered out. In the third place, the current model estimates incomes without considering the presence of houses in the images, which could lead to spurious predictions. One possible solution could be the implementation of a two-stage CNN. In the first stage, the network would identify the presence of houses within the area of interest. In the second stage, if houses are found, the network would then predict the per capita income. This strategy would not only

improve the results interpretation and precision, but also allow us to estimate better averages in each census tract, considering only the cells with houses and, thus, reducing the estimation general error. Lastly, exploring more advanced models, such as Vision Transformers or stronger CNNs, could improve the model's performance even more.

## Supporting information

**S1 Replication Scripts. GitHub repository.** Python and Stata scripts used to reproduce the results and visualizations presented in the manuscript.
(ZIP)

**S1 Fig. Gridded predictions across models.** Spatial per capita income estimates of Buenos Aires in 2013 using unstacked and stacked (50$x$50m+200$x$200m) images.
(TIF)

**S1 Table. Spatial autocorrelation across models.** Moran's I in the Autonomous City of Buenos Aires (2013) across Model Configurations and Distance Band Thresholds.
(TEX)

## Acknowledgments

This paper is a work derived from the Working Paper: Abbate, Gasparini, Gluzmann, Montes-Rojas, Sznaider and Yatche (2023), "Ingreso Estructural Por Area Geografica: una aplicación para Argentina", presented at the LVIII Annual Meeting of the Asociacion Argentina de Economía Política. We thank the Comision Nacional de Actividades Especiales (CONAE) for providing the satellite images used in this work and the Instituto Geográfico Nacional (IGN) for providing the aerial imagery used in the figures. The authors would like to express their gratitude to Ignacio Sarmiento Barbieri, Leonardo Peñaloza-Pacheco, Matias Ciasci, Ines Berniell and Cristian Bonavida for their insightful feedback and contributions to the development of this manuscript, and to Guadalupe Olmedo for her help with the translation and editing. Finally, we would like to acknowledge the support and encouragement of our families and friends during the research process. Any mistake or omission falls under our responsibility.

## Author contributions

**Conceptualization:** Nicolas Francisco Abbate, Leonardo Gasparini, Franco Ronchetti, Facundo M. Quiroga.

**Data curation:** Nicolas Francisco Abbate.

**Formal analysis:** Nicolas Francisco Abbate.

**Investigation:** Nicolas Francisco Abbate, Franco Ronchetti.

**Methodology:** Nicolas Francisco Abbate, Leonardo Gasparini, Franco Ronchetti.

**Software:** Nicolas Francisco Abbate.

**Supervision:** Leonardo Gasparini, Franco Ronchetti.

**Validation:** Nicolas Francisco Abbate, Leonardo Gasparini, Franco Ronchetti, Facundo M. Quiroga.

**Visualization:** Nicolas Francisco Abbate.

**Writing – original draft:** Nicolas Francisco Abbate.

**Writing – review & editing:** Nicolas Francisco Abbate, Leonardo Gasparini, Franco Ronchetti, Facundo M. Quiroga.

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
