## [Decision Letter · Decision Letter 0]

25 Nov 2024

PONE-D-24-42689Deep learning with satellite images enables high-resolution income estimation: a case study of Buenos AiresPLOS ONE

Dear Dr. Abbate,

Thank you for submitting your manuscript to PLOS ONE. After careful consideration, we feel that it has merit but does not fully meet PLOS ONE’s publication criteria as it currently stands. Therefore, we invite you to submit a revised version of the manuscript that addresses the points raised during the review process.

Please ensure to address specific comments on the model sensitivity to various parameters including vegetation, satellite data inherent issues such as cloud cover, temporal data influence, architechture comparisons, model calibration and validation issues.

We look forward to receiving your revised manuscript.

Kind regards,

Krishna Prasad Vadrevu, Ph.D

Academic Editor

PLOS ONE

Journal Requirements:

3. We noted in your submission details that a portion of your manuscript may have been presented or published elsewhere. “As mentioned in the cover letter, an earlier version of this manuscript has been published in L CLEI 2024 conferences. Some figures of the methodology are fairly similar, as some of the decisions from such section remain the same. Nevertheless, all the results have been updated as there where many changes in the methodology, and the results are much more presice due to these updates. Furhtermore, many subsections of results have been added as the models have been tested much further than the earlier discussion manuscript.” Please clarify whether this [conference proceeding or publication] was peer-reviewed and formally published. If this work was previously peer-reviewed and published, in the cover letter please provide the reason that this work does not constitute dual publication and should be included in the current manuscript.

5. We note that Figures 1, 2, 3, 4, 5, 6, 7, 8, 9 11, 13 and 14 in your submission contain map/satellite images which may be copyrighted. All PLOS content is published under the Creative Commons Attribution License (CC BY 4.0), which means that the manuscript, images, and Supporting Information files will be freely available online, and any third party is permitted to access, download, copy, distribute, and use these materials in any way, even commercially, with proper attribution. For these reasons, we cannot publish previously copyrighted maps or satellite images created using proprietary data, such as Google software (Google Maps, Street View, and Earth). For more information, see our copyright guidelines: http://journals.plos.org/plosone/s/licenses-and-copyright 

1. You may seek permission from the original copyright holder of Figures 1, 2, 3, 4, 5, 6, 7, 8, 9 11, 13 and 14  to publish the content specifically under the CC BY 4.0 license. 

Reviewers' comments:

Reviewer's Responses to Questions

**Comments to the Author**

1. Is the manuscript technically sound, and do the data support the conclusions?

Reviewer #1: Yes

Reviewer #2: Yes

2. Has the statistical analysis been performed appropriately and rigorously? 

Reviewer #1: Yes

Reviewer #2: No

3. Have the authors made all data underlying the findings in their manuscript fully available?

Reviewer #1: Yes

Reviewer #2: Yes

4. Is the manuscript presented in an intelligible fashion and written in standard English?

Reviewer #1: Yes

Reviewer #2: Yes

5. Review Comments to the Author

Reviewer #1: The article proposes a novel and effective application area for satellite imagery, offering significant and impactful outcomes in this context.

The EfficientNetV2 architecture was selected due to its ability to deliver high performance despite a relatively low number of parameters, making it an efficient choice. However, the lack of a detailed comparison with alternative architectures can be considered a point open to critique.

Although this high resolution allows for the creation of more detailed income maps, it should be noted that the generalization capacity of the study may be limited due to its reliance on non-public data available only for specific years.

The training, validation, and test sets were separated by considering the different characteristics of urban areas. However, low-density population regions were excluded from the analysis, which may make it challenging to generalize the results to the entire population base.

In this context, although the article presents a highly innovative method, further work is needed on generalization, model explainability, and ethical dimensions.

There is no reference from the year 2024 among the sources cited in the article. The references cover works published between 2010 and 2023. This could be considered a point to evaluate in terms of potential gaps or limitations in the coverage of the most current literature.

In terms of potential enhancements, I suggest considering alternative measures for assessing income inequality and poverty indicators. For example, Theil Index and Atkinson Index could provide additional perspectives on income inequality, capturing variations and sensitivities that may complement the Gini coefficient analysis. Furthermore, alternative poverty measures, such as Absolute and Relative Poverty Rates, Headcount Index, and Poverty Gap Index, may offer a more comprehensive view of poverty depth and distribution characteristics. Incorporating these could broaden the analytical scope and contribute to a richer interpretation of the data.

While these measures are explained in your analysis, I would recommend providing further elaboration to enhance their clarity and impact. A more detailed breakdown of how each indicator—such as the 90/10 and 50/10 Ratios for capturing disparities across income segments, the Gini Coefficient for summarizing overall inequality, and the FGT indices (FGT(0), FGT(1), and FGT(2)) for measuring poverty dimensions—uniquely contributes to understanding inequality and poverty would enrich readers' grasp of the study’s analytical depth and context.

There are no issues with the language usage or visuals in the article.

Reviewer #2: Deep learning with satellite images enables high-resolution income estimation: a case study of Buenos Aires

While the study effectively demonstrates the use of deep learning and satellite imagery for income estimation and provides valuable insights, I have some concerns that, if addressed, could strengthen the work further.

Major comments

The training relies on the census and survey data from 2010, which may not fully capture current or latest economic conditions. Since the model performance depends on similarities between training and prediction areas; fine-tuning is required for new regions. The claim that the model can predict income for different years without updated data feels like a stretch. While it might catch long-term trends, incorporating short-term factors like inflation or employment changes would likely improve accuracy over time. More justification including explanations are required on this specific aspect.

Other Uncertainties: The model seems to lean heavily on visual features for predicting income, but this could overlook key socio-economic factors like migration. This might lead to biased results, especially in similar-looking/or developing neighborhoods.

In addition, the model's sensitivity to times of day, vegetation, shadows, or the satellite’s angle suggests that external conditions can also affect model predictions. Thus, the errors associated with the image data and subsequent image processing too can affect the results. In particular, if using any temporal data, the errors can add up.

Temporal constraints: The methodology has limited ability to detect short-term economic fluctuations; the model primarily captures structural and spatial variations. It is not clear how short-term fluctuations can be captured.

Images: The authors used an approach that combines multiple image sizes (e.g., 50x50m and 200x200m) which improves context but could introduce noise. More explanation may be provided as to how the noise was reduced.

Broad results: Results are sensitive to urban patterns in Buenos Aires and might not generalize well to different cities. In essence, the model developed can be highly local. Since the model is trained on data from Buenos Aires, there’s a risk of overfitting. Authors mention that the model can be applied to other cities, but might be overlooking some critical differences between cities—like cultural, economic, or geographical factors. Including more localized training data for each region could boost the accuracy. I suggest the authors elaborate on the discussion part on how the model can be fine-tuned to apply to other regions.

Minor comments

Abstract may be clarified to include specific time periods or intervals of the study.

6. PLOS authors have the option to publish the peer review history of their article (what does this mean?). If published, this will include your full peer review and any attached files.

Reviewer #1: No

Reviewer #2: No

---

## [Author Response · Author response to Decision Letter 1]

24 Jun 2025

#############

Editorial Comments

#############

Comment:

Thank you for submitting your manuscript to PLOS ONE. After careful consideration, we feel that it has merit but does not fully meet PLOS ONE’s publication criteria as it currently stands. Therefore, we invite you to submit a revised version of the manuscript that addresses the points raised during the review process.

Reply:

Thank you for providing us with the opportunity to revise our manuscript, "Deep learning with satellite images enables high-resolution income estimation: a case study of Buenos Aires" (PONE-D-24-42689), and for the detailed feedback on journal requirements. We have carefully addressed each point as outlined below.

The main changes in the paper, compared to our previous version, are the following:

1. Restructured discussion. Former Section 7 was split into:

Section 7 – “Conceptualizing the Model’s Measurement of Income,” clarifying that the model targets permanent rather than short-term income;

Section 8 – “Generalizing the Model to Other Cities,” with new content on fine-tuning, transferability, explainability, and ethics.

2. New inequality metrics. Theil and Atkinson (ε = 1, 2) indices were added to Table 4; Section 6.3 now contains an expanded primer on all inequality / poverty indicators (90/10, 50/10, Gini, FGT family, etc.).

3. Additional robustness material. A new footnote, Figure S1, and Table S1 demonstrate how stacking 50 × 50 m and 200 × 200 m tiles reduces spatial noise; Moran’s I statistics at four distance bands are reported.

4. Limitations emphasised. New text highlights sensitivity to low-density areas, external imaging conditions, and the need for a two-stage model (household-presence → income) in sparsely populated zones.

Furthermore, some additional minor modifications where included:

5. The abstract now states the forecast years (2013, 2018, 2022).

6. The bibliography includes several 2024 studies on transformer-based remote-sensing models.

7. We have corrected a few spelling mistakes and writing inconsistencies.

8. We have added a footnote (page 2) about the relevance of the data we've employed, even if it is proprietary.

9. The MSE for the first two rows of Table 3 was corrected, as they were abnormally high due to a transcription error. Now those results are more in line with the other models.

10. We have explicitly incorporated the replication package (with the corresponding limitations) into the Supporting Information.

11. Figure 1 was updated to correct a small mistake in the order of sections 4.1 and 4.2; and Figure 6, the legend has been updated so the colors for the census tracts and the satellite images now match their intended assignments.

Comment:

Reply:

We have reviewed our manuscript and the PLOS ONE style templates and believe that the revised manuscript fully adheres to the style requirements.

Comment:

Please note that PLOS ONE has specific guidelines on code sharing for submissions in which author-generated code underpins the findings in the manuscript. In these cases, we expect all author-generated code to be made available without restrictions upon publication of the work. Please review our guidelines and ensure that your code is shared in a way that follows best practice and facilitates reproducibility and reuse.

Reply:

We understand and agree with PLOS ONE's guidelines on code sharing. All author-generated code that underpins the findings in this manuscript, including scripts for the Small Area Estimation (SAE) process, image preprocessing (such as stacking and pansharpening logic), CNN model training, and the generation of final income maps, are already available without restrictions on GitHub. We have explicitly included this repository as supporting information (S1 Replication Scripts).

Comment:

We noted in your submission details that a portion of your manuscript may have been presented or published elsewhere. “As mentioned in the cover letter, an earlier version of this manuscript has been published in L CLEI 2024 conferences. Some figures of the methodology are fairly similar, as some of the decisions from such section remain the same. Nevertheless, all the results have been updated as there where many changes in the methodology, and the results are much more presice due to these updates. Furhtermore, many subsections of results have been added as the models have been tested much further than the earlier discussion manuscript.” Please clarify whether this [conference proceeding or publication] was peer-reviewed and formally published. If this work was previously peer-reviewed and published, in the cover letter please provide the reason that this work does not constitute dual publication and should be included in the current manuscript.

Reply:

Thank you for the opportunity to clarify the previous presentation of portions of this work.

The manuscript presented at the "L CLEI 2024 conferences" underwent peer review and was formally published as conference proceedings. However, it represents an earlier, preliminary stage of the current research. Specifically, the CLEI publication, written entirely in Spanish, was an initial exploration and proof-of-concept emphasizing the feasibility of employing remote sensing data for urban income estimation. It provided only preliminary results and a foundational methodological framework. Additionally, the CLEI paper was limited in scope, comprising only 10 pages including figures.

In contrast, the manuscript submitted to PLOS ONE constitutes a substantial and novel advancement of that initial work, extending to almost 30 pages without figures, reflecting the substantial changes and advancements between both manuscripts. Key differences and enhancements include:

- Methodological Refinements: The methodology has significantly evolved, incorporating a novel multi-resolution image stacking approach (combining 50x50m and 200x200m resolutions), which was entirely absent from the CLEI paper. This innovation significantly improved model accuracy, demonstrated by an R² of 0.878.

- Expanded Dataset: The current manuscript utilizes a substantially expanded dataset, incorporating images from multiple years (2013, 2018, and 2022). In contrast, the earlier publication focused solely on imagery from 2013, limiting its temporal robustness and scope.

- New Results and In-depth Analyses: The manuscript submitted to PLOS ONE introduces extensive new findings, including detailed temporal income estimations for 2018 and 2022, comprehensive qualitative case studies, and robust validations against census and survey data at multiple spatial aggregation levels. These critical analyses and results were not included in the CLEI publication.

- Enhanced Spatial Resolution and Validation: The current manuscript provides extensive validation and thorough discussion of the high-resolution (50x50m) estimations, presenting this core contribution for the first time in detail.

- Conceptual Contributions: Additionally, the current manuscript introduces substantial new conceptual insights regarding the measurement of income through remote sensing and broader considerations for model generalization and applicability.

Given these considerable methodological advancements, significantly expanded results, and new analytical depth, we believe the submitted manuscript represents a meaningful and distinct contribution beyond the initial CLEI conference publication. We have explicitly cited the earlier conference paper within the current manuscript to ensure complete transparency.

Comment:

We note that you have indicated that there are restrictions to data sharing for this study. PLOS only allows data to be available upon request if there are legal or ethical restrictions on sharing data publicly.

b) If there are no restrictions, please upload the minimal anonymized data set necessary to replicate your study findings to a stable, public repository and provide us with the relevant URLs, DOIs, or accession numbers. You also have the option of uploading the data as Supporting Information files, but we would recommend depositing data directly to a data repository if possible.

Reply:

Thank you for highlighting this important issue regarding data availability.

The satellite imagery used in this study is proprietary and subject to legal restrictions imposed by third-party data providers, specifically Airbus DS Geo SA and the Argentine National Commission for Space Activities (CONAE). According to the end-user license agreement signed with these providers, we are not permitted to publicly share the raw satellite imagery.

Researchers interested in replicating the full analysis pipeline, starting from the original source imagery, must acquire this data commercially. The proprietary Pléiades and Pléiades NEO satellite imagery is owned by Airbus and can be purchased through their data portal: https://space-solutions.airbus.com/imagery/. To facilitate access, we provide unique product identifiers for each scene used in our study within the S1 Replication Package and S1 Replication Scripts. Researchers can use these identifiers to locate and purchase the exact imagery from Airbus.

However, we expect most researchers will primarily seek to replicate results either before model training (i.e., generating the small-area estimates) or the analytical outcomes derived from the model predictions (i.e., all analyses presented from section 6.1 onwards). To this end, we have made publicly available the necessary data and scripts for replicating these aspects of the study. These datasets are deposited in a Zenodo repository, and the corresponding replication scripts are available in a GitHub repository, both of which are cited in the Supporting Information. Specifically, we have provided:

- Argentina's 2010 Census and Survey Dataset: this dataset is used to replicate the small-area estimates employed in training our model alongside satellite images.

- Derived Income Estimates: Aggregated per capita income estimates at the census tract level (labels for CNN model training) and final 50x50m gridded income predictions for the Buenos Aires Metropolitan Area (years 2013, 2018, and 2022). These predictions enable the replication of all Figures and Tables presented in the Results section using scripts provided in the GitHub repository.

- Trained Model Weights: The EfficientNetV2 model weights (for both 4-band and 8-band inputs). Although these weights are not essential for replicating the primary findings, they are provided for further model fine-tuning in other contexts.

Comment:

We note that Figures 1, 2, 3, 4, 5, 6, 7, 8, 9 11, 13 and 14 in your submission contain map/satellite images which may be copyrighted. All PLOS content is published under the Creative Commons Attribution License (CC BY 4.0), which means that the manuscript, images, and Supporting Information files will be freely available online, and any third party is permitted to access, download, copy, distribute, and use these materials in any way, even commercially, with proper attribution. For these reasons, we cannot publish previously copyrighted maps or satellite images created using proprietary data, such as Google software (Google Maps, Street View, and Earth).

We require you to either (1) present written permission from the copyright holder to publish these figures specifically under the CC BY 4.0 license, or (2) remove the figures from your submission.

Reply:

Thank you for your comment and for bringing this important matter to our attention. To address this concern, we have reviewed all the figures and updated some of them to be licencable under CC BY 4.0.

In our paper, we have created derived figures from IGN (Instituto Geográfico Nacional de La República Argeinta) aerial images in Figures 3, 4, 6, 9, and 12. The licence of these images is attached to the submission. According to the terms and conditions outlined on their website, "Commercial use is only permitted in the case of derivative works where the information is used as an input to generate a new product." We informally consulted with IGN, and they confirmed that the updated figures in this paper qualify as "new products," making them eligible for licensing under CC-BY 4.0. We have also ensured that the appropriate attribution is mentioned in the legend of each figure. Figures 2 and 4 from the original manuscript were removed to avoid any conflicting licensing issues.

Additionally, all maps included in this paper are either originally created by the authors (Figures 1, 7, 11, and S1) or generated using delimitations provided by INDEC (Instituto Nacional de Estadísticas y Censos) (Figures 2 and 7). INDEC adheres to Argentinian Open Data law (see their disclaimer here: https://www.indec.gob.ar/indec/web/Institucional-Indec-Transparencia-1). This law (Ley N° 27275) grants users, under Article 2, the right to access, reprocess, reuse, and redistribute this information. It states, "The right of access to public information includes the ability to freely search for, access, [...], reprocess, reuse, and redistribute information held by the obligated entities listed in Article 7 of this law [...]".

Finally, Figure 5 uses a background image from UGSG/LANDSAT, which is redistributable under appropriate attribution.

#############

Reviewer # 1

#############

Comment:

The article proposes a novel and effective application area for satellite imagery, offering significant and impactful outcomes in this context.

Reply:

We would first like to thank the referee for their time and effort into crafting a very useful report. We greatly appreciate it as well as their overall positive view on our paper. In what follows we will attempt to describe how we sought to address each of their concerns into our revised manuscript.

Comment:

The EfficientNetV2 architecture was selected due to its ability to deliver high performance despite a relatively low number of parameters, making it an efficient choice. However, the lack of a detailed comparison with alternative architectures can be considered a point open to critique.

Reply:

We appreciate this comment and agree that a detailed comparison with alternative architectures could be a point of further investigation. We discussed this possibility extensively. However, we decided against conducting extensive comparative testing with additional model architectures for the following reasons:

- Focus of the Paper: Our primary aim is, besides creating high-resolution estimates for Buenos Aires, to present a replicable methodology for satellite-based income estimation using widely available data sources (census/survey data and satellite imagery), connecting the small area estimation literature from economics with the current computer vision models applications. An exhaustive comparison of specific architectures would shift focus from this core methodological contribution to a detailed applied-AI paper.

- Methodological Independence: The proposed methodology is independent of the specific vision model architecture chosen. The core contribution---linking small area estimation outputs with satellite imagery to train a predictive model---can be implemented with various suitable deep learning architectures.

- Rapidly Evolving Field: The field of computer vision models is advancing at a remarkable pace. Any ``optimal'' architecture chosen today might be superseded quickly. For instance, compared to the time when the first dr

---

## [Decision Letter · Decision Letter 1]

9 Oct 2025

PONE-D-24-42689R1Deep learning with satellite images enables high-resolution income estimation: a case study of Buenos AiresPLOS ONE

Dear Dr. Abbate,

Thank you for submitting your manuscript to PLOS ONE. After careful consideration, we feel that it has merit but does not fully meet PLOS ONE’s publication criteria as it currently stands. Therefore, we invite you to submit a revised version of the manuscript that addresses the points raised during the review process.

We look forward to receiving your revised manuscript.

Kind regards,

Beata Calka, PH.D.

Academic Editor

PLOS ONE

Journal Requirements:

Reviewers' comments:

Reviewer's Responses to Questions

**Comments to the Author**

1. If the authors have adequately addressed your comments raised in a previous round of review and you feel that this manuscript is now acceptable for publication, you may indicate that here to bypass the “Comments to the Author” section, enter your conflict of interest statement in the “Confidential to Editor” section, and submit your "Accept" recommendation.

Reviewer #2: All comments have been addressed

2. Is the manuscript technically sound, and do the data support the conclusions?

Reviewer #2: Yes

3. Has the statistical analysis been performed appropriately and rigorously? 

Reviewer #2: Yes

4. Have the authors made all data underlying the findings in their manuscript fully available?

Reviewer #2: Yes

5. Is the manuscript presented in an intelligible fashion and written in standard English?

Reviewer #2: Yes

6. Review Comments to the Author

Reviewer #2: Deep learning with satellite images enables high-resolution income estimation: a case study of Buenos Aires : round 2 review comments

The authors have made significant changes from the original submission. They have done a great job answering all the initial questions I raised, and I am satisfied with the level of detail provided. The introduction of the concept of permanent income and expanding on image noise reduction adds further clarity to the study. The addition of newer sections, especially Sections 7 and 8, strengthens the manuscript and makes it more robust and well-rounded. The authors outline a general framework with the potential to generate per capita income estimates applicable to different regions, and they clearly describe the limitations and steps needed to adapt the model across varying contexts. The inclusion of replication scripts and access to model weights is appreciated.

Within the scope of this study, focused on Buenos Aires, I am satisfied with the results. I believe the manuscript is now ready to be accepted for publication pending a few minor revisions that I outline below.

Minor Suggestions

1. In Table 3 the title "Hyperparameter Configurations" seems misleading, as the table shows high-level model settings (e.g., image size, bands, years) rather than true hyperparameters like learning rate, batch size, input, activation function or number of layers etc.. I suggest renaming it to "Model Configurations and Mean Squared Error…" or something similar.

Figures:

1. For “Figure 1”: In “6. Results” for the ‘example income deciles map built with images from 2013’, including a scale bar, north arrow and coordinates would be ideal for understanding the spatial context since this is the first introduction of the study area. If feasible, including an earlier figure that outlines the study area with map of Argentina featuring Buenos Aires would improve readability and spatial context particularly with readers that are not familiar with the region

2. For “Figure 2” – including a north arrow and the suffix for latitude and longitude would make the map more visually appealing

3. I suggest using a consistent symbol for meters throughout the manuscript. As is stands, ‘m’ is used in the manuscript. However ‘mts’ is used in Figure 3 and parts of manuscript interchangeably.

4. “Figure 5” should include a scalebar, north arrow, and coordinates suffixes. In essence, each image must be self explantory on its owns

5. “Figure 7” “a” seems cut off at the bottom or is a text label that is missing? Inclusion of scalebar, coordinates, and suffixes would be help to compare images side by side.

6. “Figure 10” I suggest including R2 value in figure caption or image subplots

7. PLOS authors have the option to publish the peer review history of their article (what does this mean?). If published, this will include your full peer review and any attached files.

Reviewer #2: No

---

## [Author Response · Author response to Decision Letter 2]

15 Nov 2025

Manuscript ID: PONE-D-24-42689

Title: Deep learning with satellite images enables high-resolution income estimation: a case study of Buenos Aires

We sincerely thank the editor and reviewers for their time and valuable feedback. We were pleased to see that the reviewer found our revisions to be a great job and was satisfied with the level of detail provided. We have addressed the final minor suggestions as detailed below.

Reviewer #2

Reviewer Comment 1: In Table 3 the title "Hyperparameter Configurations" seems misleading... I suggest renaming it to "Model Configurations and Mean Squared Error..." or something similar.

Our Response: We thank the reviewer for this excellent suggestion. The original title was indeed not precise. We have revised the title of Table 3 to “Model Configurations and Mean Squared Error for Model Selection in Test Set” to more accurately reflect its contents.

Reviewer Comment 2: For “Figure 1”: In “6. Results” for the ‘example income deciles map...’, including a scale bar, north arrow and coordinates would be ideal... If feasible, including an earlier figure that outlines the study area... would improve readability...

Our Response: We agree completely that an earlier study area map would improve spatial context for the reader. We have created a new map of the study area, which is now included as the new Figure 1. This map contains a scale bar, north arrow, and coordinates as requested. The original flowchart has been renumbered to Figure 2.

Reviewer Comment 3: For “Figure 2” – including a north arrow and the suffix for latitude and longitude would make the map more visually appealing.

Our Response: Thank you for this suggestion. The figure (now renumbered) and all other maps have been updated to include north arrows and standard coordinate suffixes (e.g., °S, °W).

Reviewer Comment 4: I suggest using a consistent symbol for meters throughout the manuscript. As is stands, ‘m’ is used in the manuscript. However ‘mts’ is used in Figure 3 and parts of manuscript interchangeably.

Our Response: This is an important point for consistency. We have reviewed the entire manuscript and all figures and have standardized the unit for meters to 'm' throughout.

Reviewer Comment 5: “Figure 5” should include a scalebar, north arrow, and coordinates suffixes... “Figure 7” “a” seems cut off at the bottom... Inclusion of scalebar, coordinates, and suffixes would be help...

Our Response: We appreciate the reviewer pointing these out. All relevant figures have been updated to be self-explanatory with the inclusion of scale bars, north arrows, and coordinate suffixes. We have also corrected the formatting issue with the label in Figure 7 (now Figure 8). The blank region was not a missing label but an area with no available satellite imagery for that year; we have clarified this in the figure caption.

Reviewer Comment 6: “Figure 10” I suggest including R2 value in figure caption or image subplots.

Our Response: This is a helpful suggestion to improve clarity. We have now included the R² values directly in each of the subplots in Figure 10.

---

## [Editor Report · Decision Letter 2]

19 Nov 2025

Deep learning with satellite images enables high-resolution income estimation: a case study of Buenos Aires

PONE-D-24-42689R2

Dear Dr. Abbate,

We’re pleased to inform you that your manuscript has been judged scientifically suitable for publication and will be formally accepted for publication once it meets all outstanding technical requirements.

Kind regards,

Beata Calka, PH.D.

Academic Editor

PLOS ONE

---

## [Editor Report · Acceptance letter]

PONE-D-24-42689R2

PLOS One

Dear Dr. Abbate,

I'm pleased to inform you that your manuscript has been deemed suitable for publication in PLOS One. Congratulations! Your manuscript is now being handed over to our production team.

Kind regards,

on behalf of

Dr. Beata Calka

Academic Editor

PLOS One